# Feasibility of Using a Type I IFN-Based Non-Animal Approach to Predict Vaccine Efficacy and Safety Profiles

**DOI:** 10.3390/vaccines12060583

**Published:** 2024-05-27

**Authors:** Hanin Abdel-Haq

**Affiliations:** Istituto Superiore di Sanità, Viale Regina Elena, 299, 00161 Rome, Italy; hanin.abdelhaq@iss.it

**Keywords:** interferon, type I IFN, type 1 IFN-related biomarkers, ISGs, non-animal approaches, biomarkers predictive of vaccine efficacy and safety, vaccine outcome profile, 3Rs-based models, systems vaccinology

## Abstract

Animal-based tests are used for the control of vaccine quality. However, because highly purified and safe vaccines are now available, alternative approaches that can replace or reduce animal use for the assessment of vaccine outcomes must be established. In vitro tests for vaccine quality control exist and have already been implemented. However, these tests are specifically designed for some next-generation vaccines, and this makes them not readily available for testing other vaccines. Therefore, universal non-animal tests are still needed. Specific signatures of the innate immune response could represent a promising approach to predict the outcome of vaccines by non-animal methods. Type I interferons (IFNs) have multiple immunomodulatory activities, which are exerted through effectors called interferon stimulated genes (ISGs), and are one of the most important immune signatures that might provide potential candidate molecular biomarkers for this purpose. This paper will mainly examine if this idea might be feasible by analyzing all relevant published studies that have provided type I IFN-related biomarkers for evaluating the safety and efficacy profiles of vaccines using an advanced transcriptomic approach as an alternative to the animal methods. Results revealed that such an approach could potentially provide biomarkers predictive of vaccine outcomes after addressing some limitations.

## 1. Introduction

Animal-based tests are used for the quality control and outcome characterization of vaccines. However, because highly purified and safe vaccines are now available, the role of animal testing should change. This change is also necessary considering that animals do not reflect many aspects of the human immune response being very complex, and the scientific validity and translatability of the obtained results to humans are questionable [1,2,3,4,5]. Additionally, there is a solicited recommendation by various international organizations to reduce or even replace animal use with in vitro methods for the characterization of established vaccines [6,7].

The European Pharmacopoeia (Ph. Eur. 5.2.14) has recognized that in comparison to in vitro methods, in vivo bioassays are time-consuming, expensive and laborious. They do not necessarily predict the actual responses in the target population and are characterized by low precision and high variability, as well as difficulty in supplying and maintaining animals. Additionally, most of the in vivo assays were established before the ICH Q2 (R1) (International Conference on Harmonisation of Technical Requirements for Registration of Pharmaceuticals for Human Use (Validation of analytical procedures: text and methodology)) or VICH GL2 (International Cooperation on Harmonisation of Technical Requirements for Registration of Veterinary Medicinal Products (Validation methods)) guidelines, and therefore they are not validated, being deemed compendial [6,7]. Furthermore, the in vivo animal model is poorly predictive of human responses and can only partially reflect antigen–cell interactions due to the presence of genetic and environmental differences between animals and humans [4,5,8,9]. In addition, the inflammatory response observed in animals following infection or vaccination presents many incompatibilities with that observed in humans [4,10,11]. On the other hand, the availability of biomarkers within non-animal models capable of predicting the vaccine outcome in terms of immunogenicity, efficacy and safety would reduce the vaccine production and testing costs [12]. Additionally, they would shorten the vaccine production time and increase global access due to the reduced costs [7]. Table 1 shows the main advantages of using non-animal methods over animal bioassays as models to assess vaccines outcomes.

Therefore, alternative 3Rs (replacement, reduction and refinement)-based approaches or strategies that are more relevant than animals for the human immune system and that can replace or reduce animal use for the assessment of vaccine outcome profiles must be established.

In vitro tests for vaccine quality control exist and have already been implemented. This is the case of two human vaccines, Bexsero^®^, the meningococcal vaccine [13], and Gardasil^®^, the human papilloma virus vaccine [14]. For both vaccines, the safety and efficacy profiles are accurately and robustly controlled using in vitro methods only. Another example is the case of the polysaccharide conjugate vaccine, the Haemophilus influenza type b vaccine, for which the use of an enzyme-linked immunosorbent assay (ELISA) during the different manufacturing steps enabled the characterization and quality control of vaccine samples [15].

Unfortunately, existing tests are specifically designed for some next-generation vaccines, and this makes them not readily available for testing other vaccines. For this reason, universal non-animal tests capable of controlling the quality of a large number and variety of vaccines are still needed.

Signatures of the immune response, such as B-cell growth factor TNFRS17, CD38, kinase CaMKIV, CXCL10, IL-6 and interferon (IFN), among others [16,17,18], could represent a promising approach to predicting vaccine outcomes using non-animal methods. In various studies, type I IFNs were found to be one of the most relevant immune signatures for this purpose. Indeed, type I IFNs play an important role in the vaccine-induced early innate immune response, being essential to elicit immune responses against acute infections, and are involved in the later adaptive immunity, being essential for the maintenance and functionality of central immune cells [17,19,20]. Additionally, several studies consider type I IFN signaling a reliable indicator of an effective vaccine response [16,21]. Furthermore, using transcriptomic approaches, type I IFN-related molecular signatures, including interferon-stimulated genes (ISGs), were found to be upregulated after vaccination with various vaccines, including yellow fever, pertussis, enterovirus type 71 (EV71) and influenza, and have been shown to be able to predict the immunogenicity of these vaccines both in terms of antibody titer and B and T cells responses [16,17,18,19,20,22,23,24,25,26,27,28,29,30]. Finally, the expression level of most of the identified type I ISGs was significantly and positively correlated with either the antibody response magnitude [16,17,19,20,22,31] or the results of specific tests for vaccine quality control performed in animal models [25,28].

On the other hand, vaccines are often associated with adverse effects (AEs), which could be severe [32,33]. Type I IFNs also, in this case, have been shown to be able to provide safety biomarkers for vaccine assessment since they are also involved in vaccine reactogenicity [34]. In fact, the unbalanced production of type I IFNs in the presence of chronic viral and bacterial infections can cause serious AEs and lead to severe inflammatory disorders and autoimmunity (reviewed in [35,36]). Overall, this evidence suggests that type I IFN responses in the context of vaccines may provide potential candidate molecular biomarkers that can predict vaccine outcome associated with type I IFN signaling in non-animal approaches.

This review will examine if this idea might be feasible by analyzing all relevant published studies that have provided type I IFN-related biomarkers for the evaluation of the safety and efficacy profiles of vaccines using advanced transcriptomic approaches as an alternative to animal models. Since this paper will not take into consideration all vaccine platforms but only the vaccines that have been tested in the studies analyzed in this paper, the suitability of type I IFN-related biomarkers to assess outcome profiles of other types of vaccines will be examined and discussed. Furthermore, a future perspective on how this approach could be applied to other types of vaccines will also be discussed.

## 2. Interferons (IFNs)

IFNs are a group of cytokines that can elicit antiviral, immunomodulatory, antiangiogenic, antiproliferative and antitumor effects. IFNs are released by cells following exposure to various stimuli, including viruses and double- and single-stranded nucleic acids [37]. IFNs can be distinguished as type I (IFN-α, IFN-β, IFN-δ, IFN-ε, IFN-ζ, IFN-κ, IFN-μ, IFN-ν, IFN-τ and IFN-ω, which all bind to IFNAR), type II (IFNγ, which binds to IFNGR) and type III (IFNλ1-4, which binds to IFNLR) [37,38]. Very recently, a type IV IFN (IFN-υ) has been identified and characterized by Chen and colleagues [39]. This new IFN type is expressed in vertebrates, such as zebrafish (Danio rerio) and African clawed frogs (Xenopus laevis), but not in humans. It binds a class II cytokine receptor complex (IFN-υR1 and IL-10R2/CRFB4), the stimulation of which leads to the activation of antiviral responses and the induction of ISGs [such as *grass carp reovirus (GCRV)-induced gene 2 (Gig2)*, *interferon regulatory factor 1 (IRF1)*, *myxovirus resistance protein A (MXA)* and *radical S-adenosyl methionine domain-containing 2 (RSAD2)*, also known as *virus-inhibitory protein*, *endoplasmic reticulum-associated*, *IFN-inducible (Viperin)*], in both zebrafish and African clawed frogs [39]. Like other IFNs, the type IV IFN has ~200 amino acids and ~6 alpha helices, while in contrast to other IFNs, the sequence and structure of the N-terminal ligand-binding chain of the IFN-υ are different [39].

Figure 1 shows a schematic design of all IFNs and their receptors and signaling pathways that have been identified and characterized to date.

Since much of the available information on IFNs had already been covered by several outstanding reviews, this paper will only consider type I IFNs, signaling pathways and ISGs as the central focus of this study.

### 2.1. Type I IFNs

Type I IFNs include a major number of subtypes, of which the IFN-α and IFN-β subtypes are the best characterized. IFN-α has more than 13 subspecies and is predominantly secreted by infected cells and hematopoietic cells, in particular by natural IFN-α-producing cells (IPCs), also known as plasmacytoid dendritic cells (pDCs), and is mainly present in the bone marrow and spleen [40], whereas IFN-β is ubiquitously produced by immune cells and fibroblasts [41]. A recent study revealed that macrophages, monocytes and conventional dendritic cells (cDCs) are the main sources of type I IFNs, in addition to pDCs, after infection with viruses, bacteria, fungi and eukaryotic parasites in in vivo models (reviewed in [42]). Another previous study classified the type I IFN sources based on the compartment involved in the infection; for example, pDCs, marginal zone macrophages, monocytes and non-hematopoietic stromal cells were considered IFN primary sources in some systemic viral infections; epithelial cells and alveolar macrophages were considered IFN primary sources in respiratory viral infections; parenchymal cells, fibroblasts, tissue resident macrophages and DCs were considered IFN primary sources in infected organs; subcapsular sinus macrophages were considered IFN primary sources in draining lymph nodes; and epithelial cells, fibroblasts, tissue-resident macrophages and DCs were considered IFN primary sources in skin and mucosal infections (reviewed in [43]). Such a variety of type I IFN sources was also confirmed in tumor-related studies conducted in chimeric mice [44]. The production of type I IFNs from these multiple sources does not occur stochastically but in a coordinated and cooperative manner and is also necessary for an efficient anti-infective response during all stages of infection. pDCs, macrophages and DCs are mainly essential in the initial phase of the infection, while the other cell types become involved as a source of IFNs in the later stages [42,44]. Findings by Kumagai et al. suggested that in certain tissues and infection conditions, pDCs start to produce type I IFNs only when other major sources of type I IFNs are impaired [45]. pDC-, macrophage- and DC-derived type I IFNs are essential to elicit antiviral innate responses against acute infection and can promote adaptive immunity by inducing the differentiation of B cells into plasmablasts, which mature into antibody-secreting plasma cells [46,47,48], and preventing death of activated T cells during infections [49]. Furthermore, type I IFNs play central roles in the differentiation, proliferation, maturation, survival or activation of immune cells crucial for innate immunity, such as natural killer (NK) cells, monocytes and DCs [48]. Type I IFNs have been shown to be able to enhance the antibody response, stimulate the secretion of immunoglobulin G (IgG) and A (IgA) and induce the sustained production of long-lived antibody and memory B cells. Thus, type I IFNs act as direct effectors of innate immunity and as promoters of the adaptive immune response.

All type I IFN subtypes are structurally similar and explicate their functions through a heterodimeric transmembrane receptor belonging to the class II helical cytokine receptor family [50] and composed of two glycosylated subunits called IFN-α receptor 1 (IFNAR1) and IFN-α receptor 2 (IFNAR2a,b,c) [37]. Due to this peculiarity, type I IFN subtypes share common behaviors and characteristics. Increasing evidence indicates that each individual type I IFN subtype has a distinguished, cell type-specific and stage-dependent antiviral function [51,52,53]. For example, IFN-α acts during the early stage of viral infection, representing an essential modulatory activity of the host’s immune response, while IFN-β acts at a later stage of viral infection, preventing the progression of inflammation [51,52]. Fox et al. attributed the specialization of type I IFN subtypes to the host’s environment and its interactions [53], along with the distinctive properties of these subtypes themselves.

### 2.2. Type I IFN Signaling Pathways

Type I IFNs are secreted upon the activation of innate immune signaling pathways via pattern recognition receptors (PRRs), such as toll-like receptor (TLR), C-type lectin receptor (CLR), *retinoic acid-inducible gene I (RIG-I)*, Nod-like receptors (NLRs) and cytosolic DNA/RNA sensors, which signal via the *stimulator of interferon genes (STINGs)* [54]. Once activated, PRRs lead to the activation of nuclear transcription factors, including interferon regulatory factor 3 (IRF-3) and 7 (IRF-7) and nuclear factor κB (NF-κB). Activated NF-κB exerts a negative regulatory effect against IFNs and their signaling pathway [55], while instead, IRF-3 (constantly and ubiquitously expressed) and IRF-7 (sparsely expressed in the absence of infection) together induce the immediate production of type I IFNs (IFN-α (subtype 4) and IFN-β) and elicit an effective antiviral action. IRF7 is essential for the expression of type I IFNs throughout all phases of viral infection, and therefore, the lack of IRF7 facilitates viral infection [56] and in some cases of severe influenza is even life-threatening [57,58]. IRF7, on the other hand, having a brief duration of action (0.5–1 h), impedes the prolonged expression of IFNs, thus preventing an eventual toxic effect against the host cells [56]. IRF3 is constitutively and abundantly expressed and exerts its antiviral effect during the early phase of viral infection [56]. IRF3 promotes the further production of type I IFNs (mainly IFN-β) either directly or indirectly, for example by increasing the expression levels of IRF7 through IFN-β-mediated induction [56,59]. IRF3, unlike IFR7, is degraded rapidly after infection without being newly reproduced, and like IRF7, the lack of IRF3 can negatively affect the immune response due to the lack of induction of IFN-α/β [60]. In relation, a recent study revealed that IRF3 can facilitate virus latency during chronic infections and that this role is influenced by the inoculum infectious load and the route of virus transmission [61]. The reason for this IRF3-mediated virus latency is unclear because multiple factors and interactions between the virus and the immune system are involved, based on the interpretation provided by the authors of that study. One of the explanations provided by these authors is that this virus latency could be related to the reduction in the expression of key antiviral ISGs in the absence of IRF3. Additionally, these authors highlighted that type I IFNs are not the only IFNs involved in chronic infection. On the other hand, IRF3 has been shown to be a weak inducer of IFN expression in the absence of IRF7. This suggests that IRF7 may contribute to the observed IRF3-induced virus latency when its level is insufficient to induce sufficient levels of IFN-α/β [56,62]. It has been demonstrated that IRF7 has a central role in maintaining and amplifying the production of high levels of IFNs during the advanced phase of the viral infection [60] and that this role is relevant against virus latency and reactivation during chronic infections [63].

Type I IFN receptor subunits IFNAR1 and IFNAR2 are associated with tyrosine kinase 2 (TYK2) and Janus kinase 1 (JAK1), respectively [64]. When a type I IFN binds to IFNAR1 and IFNAR2, two signaling pathways can be distinguished: the classical pathway (canonical pathway) and the non-classical pathway (non-canonical pathway) [64,65].

The classical type I IFN pathway activates JAK-STAT signaling, thus leading to the transcription of ISGs through a distinct mechanism (Figure 1B). Indeed, type I IFN receptor stimulation leads first to auto-phosphorylation and then to the cross-phosphorylation and activation of JAK1 and TYK2 (two non-receptor tyrosine kinases members of JAK family, which also includes JAK2 and JAK3 members). Subsequently, activated JAKs lead to the phosphorylation of the receptor subunits and the formation of several binding sites for the recruitment of STAT1 and STAT2. Then, phosphorylated STAT1 and STAT2 proteins, after their dimerization, translocation to the nucleus and binding to IRF9, lead to the formation of the IFN-stimulated gene factor 3 (ISGF3) complex. ISGF3 binds to DNA sequences called IFN-stimulated response elements (ISREs) and leads to the transcription of ISGs [64,66,67].

Type I IFNs can also signal in a non-classical mode, which is mediated by unphosphorylated STAT3. Using a genome-wide transcriptional analysis approach in Drosophila, Tsurumi and colleagues found that some overlap can occur between the classical and non-classical JAK-STAT pathways, although the two pathways regulate distinct target genes [68].

The non-classical type I IFN signaling pathways mainly include the mitogen-activated protein kinase (MAPK) pathway, the phosphatidylinositol 3-kinase (PI3K)/mammalian target of rapamycin (mTOR) pathway and the guanine exchange factor (GEF) pathway. These pathways can elicit effects on the mRNA translation and expression of ISGs and interact with STATs of the canonical pathways [65]. This is the case of the Schlafen (SLFN) proteins, which are implicated in important biological activities related to the immune defense against viral agents. After IFNα treatment of embryonic fibroblasts derived from knockout mice, several SLFN members were found to be upregulated, and both STAT1 and STAT3 were essential for the induction of all SLFN genes except the SLFN5 gene (this missed induction of SLFN5 refers to STAT3 only) [69,70]. Additionally, there is evidence that some members of the SLFN protein family may be induced and regulated by the same type I IFN upstream effectors that regulate ISGs of classical signaling, such as *ISG15* [70]. These examples clearly highlight the incomplete independence of non-classical signaling from classical signaling.

Strong evidence demonstrates that type I IFN JAK-STAT signaling can be sustained by positive feedback regulation. This positive feedback is believed to be triggered by a reduction in the levels of phosphorylated STAT proteins activated by the continuous production of type I IFN under particular conditions of infection [71].

Type I IFN JAK-STAT signaling can also be sustained by direct regulation mediated by some endogenous molecules, such as the ubiquitination regulatory X (UBX) domain-containing protein 6 (UBXN6). Ketkar and colleagues revealed that an increase in UBXN6 levels is strongly correlated with an increase in type I IFN production and prolonged ISG expression or vice versa [72]. UBXN6 seems to explicate this regulatory function by directly binding TYK2 and inhibiting the IFN-β-induced degradation of TYK2 itself and IFNAR1 [72].

JAK-STAT signaling can also be inhibited by negative feedback regulation, which can be explicated in several ways. For example, the suppressor of cytokine signaling 1 (SOCS1), the best-known negative regulator of JAK-STAT signaling, dampens type I IFN signaling through a selective interaction with the IFNAR1-associated TYK2 and the inhibition of TYK2-mediated STAT signaling [73]. The tripartite motif-containing-10 (TRIM-10) attenuates type I IFN-JAK-STAT signaling by binding IFNAR1 and inhibiting the association of IFNAR1 with TYK2 [74]. On the other hand, ISG ubiquitin-specific peptidase 18 (USP18 or UBP43), inhibits type I IFN-JAK-STAT signaling through specific and competitive binding of the USP18 to IFNAR2 to prevent the interaction between IFNAR2 and JAK1 [75,76]. USP18 specifically regulates type I IFN signaling, although it is induced after stimulation with both type I and type III IFNs. Recently, Dagenais-Lussier et al. showed that USP18 overexpression in human memory CD4 T cells from immunodeficiency virus type 1 (HIV-1)-infected patients has no impact on type I IFN signaling, which remained elevated in these cells [77]. These authors also showed that USP18 overexpression in HIV-1-derived memory CD4 T cells contributes to the loss of memory CD4 T cells themselves and other HIV-1-specific cells during the early phase of HIV-1 infection [77]. Since memory CD4 T cells are essential to confer lasting immunity in the presence of persistent viral infections, such as that caused by HIV-1, their loss leads to a weakened immune response [77]. Based on these interesting findings, USP18 can attenuate the immune response even without dampening the type I IFN signaling.

Finally, it has been reported that STAT1 transcriptional activity and NF-κB signaling activation, which regulate both the transcription and expression of ISGs, can be inhibited by SLFN5 and SLFN2, respectively [78,79].

### 2.3. Interferon-Stimulated Genes (ISGs)

Hundreds of ISGs are induced following the binding of type I IFNs to the IFNAR. However, not all induced ISGs are involved in antiviral defense, which covers both innate and adaptive immunity [54,80].

ISGs act on different levels of the viral life cycle, as schematically illustrated in Figure 2.

For example, IFN double-stranded RNA (dsRNA)-dependent protein kinase R (PKR), which can promote IFN production and further ISG induction, mediates antiviral and proapoptotic activities [81]. In relation to this activity, PKR induction has been shown to be associated with IRF3 activation and increases in the levels of IFN-β [82]. PKR additionally mediates an overall translational inhibition of both viral and cellular mRNA through activation by the phosphorylation of the α-subunit of eukaryotic translation initiation factor 2-alpha (eIF-2α) [81]. 2′–5′oligoadenilate synthetase (OAS) proteins, primarily OAS3, contribute to the inhibition of protein synthesis and viral replication by inducing the enzymatic degradation of host and viral dsRNA through the activation of RNAse L (a ubiquitous cellular endoribonuclease RNase L) [83,84]. The IFN-inducible transmembrane protein (IFITM) family is composed of several members; however, only IFITM-1, IFITM-2 and IFITM-3 are induced by IFNs (type I and type II). IFTIM proteins have a restriction activity that can prevent the penetration of different kinds of enveloped and non-enveloped viruses through the host cytoplasmic membrane. The IFTIMs’ restriction activity is primarily exerted during the late step of the viral endocytic process and could be impacted by the mechanism of entry (for example, via endocytosis or direct fusion with the plasma membrane) of every single virus through the host membrane [85]. IFTIM family proteins can inhibit virus replication, but this is likely possible during the early stage (before or during endocytic entry or fusion with the host plasma membrane by the virus) rather than during the late stage of the viral replication cycle [85,86]. dsRNA-specific adenosine deaminase acting on RNA 1 (ADAR) is an enzyme that can disturb the nucleic acid base sequential order and can also protect the host cells from virus-mediated cytotoxicity. ADAR causes the destabilization of dsRNA by promoting the deamination of adenosine and consequent formation of inosine in dsRNA [81]. Accumulating evidence indicates that ADAR also has pro-viral activity, which is realized through the inhibition of PKR phosphorylation activity and is reflected in increased host susceptibility to viral infections [87,88,89,90]. MX proteins (MXA (human)/MX1 (mouse) and MXB/MX2) have broad antiviral activities, which are mainly directed against the entry of essential viral components into the nucleus, where viral transcription and replication occurs. Human MXA, being localized to the cytoplasm, inhibits virus replication immediately after virus entry into the cytoplasm by binding and inactivating viral nucleocapsids, thus impeding the virus from reaching the nucleus [91]. While human MXB, being located very close to the nuclear pores of the outer nuclear membrane, impedes the entry of both the virus and its viral content into the nucleus through direct interaction with the viral capsid [91,92]. RSAD2 or Viperin has a broad spectrum of antiviral actions against different types of viruses, such as influenza A virus, hepatitis C virus, multiple flaviviruses, Japanese encephalitis virus and others [93]. Viperin realizes its antiviral activity through different mechanisms; for example, it can induce the degradation of viral proteins, which is likely dependent on the proteasome of viral proteins and virus species [94]. Additionally, viperin can restrict virus replication in determined regions and cell types of the central nervous system (CNS) [95]. It has recently been shown that viperin restricts viral replication by inhibiting RNA translation by inducing ribosomal collisions and activating a specific signaling pathway [96]. Furthermore, viperin can restrict virus release from infected host cells by altering membrane fluidity as a result of the disruption of lipid rafts [97,98]. Viperin has recently been shown to be able to restrict the rotavirus release from infected host cells without significantly affecting its replication by binding to a rotaviral non-structural protein 4 (NSP4) with subsequent translocation of the complex viperin–NSP4 from the endoplasmic reticulum (ER) to the mitochondria, wherein viperin entraps NSP4 and leads to impairment of the NSP4-activated intrinsic apoptotic pathway, which is essential for rotavirus release [99]. An important type I IFN-induced ISG is *ISG15* or *UCRP*, which can act directly, inhibiting viral replication (reviewed in [100]), or indirectly, activating PKR, which can inhibit the translation of cellular and viral proteins [101]. Also, human *ISG15* can influence type I IFN signaling through a regulatory function towards IRF3 and USP18. Indeed, *ISG15* can sustain the type I IFN pathway by increasing the stability of IRF3 [102], which consequently leads to the further production of type I IFNs and expression of ISGs. On the other hand, *ISG15* can attenuate type I IFN signaling via USP18 stabilization. This stabilization is achieved through the binding of I*SG15* to USP18. Then, this binding prevents the degradation of USP18 and maintains its functional levels [103]. Since USP18 is considered a key negative regulator of type I IFN signaling, reduction in its levels due to the absence or reduction in the levels of *ISG15* allows for the continuous production of type I IFNs, which can lead to the persistence of the inflammatory response (autoinflammation) [103]. Another important ISG is the *lymphocyte antigen 6 complex*, *locus E (LY6E)*. *LY6E* is involved in the activation and maturation of T and B immune cells [104]. Schoggins et al., after screening of an array of ISGs whose expression and consequent antiviral activity were induced in various cell cultures by different well-characterized viruses, including HIV-1, found that *LY6E* consistently increased viral replication both in terms of frequency and number of infected cells [88]. In line with this finding, results from other laboratories demonstrated that *LY6E* is highly expressed in human monocytes (CD4+ T cells) and primary peripheral blood mononuclear cells (PBMCs) and that this expression is significantly increased in chronic HIV-1 infections. Additionally, increased HIV-1 entry and replication were observed in these cells, which are characterized by expressing high levels of CD4 [105,106]. In contrast, Yu et al. recently demonstrated that *LY6E* inhibited HIV-1 penetration and replication, likely by facilitating the internalization of the viral receptor CD4 into human monocyte-derived macrophages and Jurkat cells, which express low levels of CD4 [107]. These findings clearly suggest that *LY6E* has a modulatory capacity, which can result in antiviral or proviral activity based on the localization on the membrane and the expression levels of determined cell surface molecules, such as the viral receptor CD4, which are necessary for some viruses, such as HIV-1, to enter target cells. This bidirectional modulation by *LY6E* also depends on the type of cell in which these molecules are expressed [105].

## 3. Potential Type 1 IFN Correlates of Vaccine Immune Response Efficacy and Safety That Might Be Used in the Future as Alternatives to Animal Models

Systems vaccinology is an emerging field that applies systems biology approaches and predictive models to vaccinology and provides a powerful tool for studying vaccine immune response outcomes [108]. Over the past decade, several research groups have increasingly and successfully used this promising approach to identify molecular biomarkers to evaluate the efficacy and safety of vaccine immune responses associated with type I IFN signaling, as detailed below and illustrated in Figure 3 and Figure 4.

### 3.1. Methods, Assays and Systems Used to Identify Biomarkers for Vaccine Efficacy

Investigated vaccines and adjuvants: live attenuated yellow fever vaccine (YF17D), whole-virus inactivated enterovirus type 71 (EV71) vaccine, 23-valent polysaccharide pneumococcal vaccine, live attenuated influenza vaccine (LAIV) and trivalent inactivated influenza vaccine (TIV) with and without the MF59 adjuvant.

Vaccine efficacy assessment and biomarker identification were performed using whole blood and serum from human vaccinees, PBMCs isolated from the blood of human vaccinees and/or human PBMCs stimulated with vaccines. Gene expression analysis was performed by quantitative real-time polymerase chain reaction (RT-qPCR). Transcriptomic analysis was performed by microarray transcriptional profiling. Vaccine-induced cytokines were analyzed by multiplex cytokine analysis. Plasmablast, memory B-cell or T-cell responses were assessed by multi-parameter flow cytometric analysis or ELISpot analysis. Hemagglutinin inhibition and virus neutralization assays were performed by a cytopathic effect test or plaque reduction neutralization test. Neutralizing antibody titers were measured by enzyme-linked immunosorbent assay (ELISA). Pathway enrichment analysis was performed through the consultation of a gene ontology (GO) database (http://www.geneontology.org) or whole-genome transcriptional profiles.

Data from microarray gene expression and gene ontology were analyzed and profiled using the open source Bioconductor platform (using the R software package V. 3.4.1), CRAN package fastICA, ingenuity pathway analysis (IPA) software and the DAVID bioinformatics database (http://david.abcc.ncifcrf.gov/) for associated pathway confirmation; GeneSpring (http://www.chem.agilent.com/en-s/products/software/lifesciencesinformatics/genespringgx/) for standard correlation with average linkage hierarchical clustering analysis; TOUCAN (http://homes.esat.kuleuven.be/~saerts/software/toucan.php) for transcription factor binding site (TFBS) analysis; classification methods called classification to nearest centroid (ClaNC)15 and discriminant analysis via mixed integer programming (DAMIP) to predict the magnitude of the CD8+ T-cell population; modular analysis of changes in transcript abundance and/or logistic regression analysis for the correlation of gene expression; antibody titer data and cross-validation prediction analysis.

### 3.2. Correlates of Vaccine Efficacy

Two studies performed separately by Querec et al. and Gaucher et al. were the first to test systems biology approaches and profile genes involved in innate and adaptive immune responses to the yellow fever vaccine (Table 2). The results obtained from these two independent studies showed that the immune response to the live attenuated yellow fever vaccine is preceded by an orchestrated overexpression of key transcription factors, including STAT1 and IRF7, which can elicit a broad and long-lasting immune response [16,22]. Also, the results from both studies led to the identification of potential specific molecular correlates of vaccine efficacy and highlighted type I IFN-related genes as being among the most promising potential correlates of protection. Indeed, in both studies, several type I IFN-induced genes (Table 2) were highly overexpressed with a peak change for most genes reached on day 7 after vaccination. These genes were also correlated with the antibody response and accompanied by an upregulation of other vaccine-induced genes associated with the complement, macrophages, NK cells or B cells.

Subsequently, using multiple novel transcriptional profiling approaches and advanced computational analyses that allow for the simultaneous and accurate analysis of all genes and associated pathways, systems biology was applied to other vaccines, such as EV71, pneumococcal and influenza. The results of these studies confirmed the predominant overexpression of type I IFN-related genes in either in vivo/ex vivo or in vitro studies, as shown in Figure 3 and Table 2.

Specifically, 220 genes were found to be significantly upregulated in PBMCs obtained from the blood of children vaccinated with the EV71 vaccine. The transcriptional profile of these 220 genes revealed the activation of *IRF7*, among other key type I IFNs and antiviral genes, both during the first (primary) and second (booster) immune responses to the vaccine [23].

Comparative analysis of the upregulated genes between the first and second immune responses revealed the presence of 124 genes common between the two responses. These common genes were more expressed in the second than in the first response. Furthermore, genes involved in inflammatory and cellular and humoral immune responses were found to be upregulated only in the second response [23]. Considering that the primary response confers a pre-immunization state while the recall response is characterized by a persistent production of neutralizing antibodies, which are known to be associated with long-term protective immunity [112], and there is robust evidence of the crucial role of *IRF7* in regulating both innate and adaptive immune responses [113], these results suggest that the activation of type I IFN signaling may be essential to achieve the desired protective immune response elicited by the EV71 vaccine. In line with this suggestion, an increase in the type I IFN levels after stimulation of the cotton rat osteosarcoma cell line in vitro and cotton rats in vivo with inactivated measles virus allowed for the further production of anti-measles antibodies and especially the achievement of the necessary protection against infections [114].

Among the upregulated type I IFN-related genes, *MX1*, which has a specific antiviral action against many viral infections, was identified as a potential marker to predict the immune persistence of the EV71 vaccine, since its expression level in the recall response was strongly correlated with the levels of the titers of EV71 neutralizing antibody at 180 days post-vaccination [23].

Comparisons between a trivalent inactivated seasonal influenza vaccine and a 23-valent pneumococcal vaccine showed that these two vaccines elicit in whole blood a distinct transcriptional signature involved in both innate and adaptive immunity [20]. Both vaccines induced robust responses on day 1 post-vaccination, but on subsequent days (7 and 10), almost only the pneumococcal vaccination produced a consistent response. Additionally, 15 h post-vaccination, the pneumococcal vaccine increased the transcription of myeloid- and inflammation-related genes, whereas the influenza vaccine induced the transcription of IFN-related genes (Table 2) in a manner directly proportional to the magnitude of the antibody response and vaccine efficacy. This difference between the responses of the two vaccines is due to their different compositions, which are made from inactivated viral particles for the seasonal influenza vaccine and a mix of polysaccharides for the pneumococcal vaccine [20].

The early induction of IFN-related genes by the influenza vaccine was previously observed by Bucasas and colleagues [19]. In that study, the authors revealed that among 494 genes found to be overexpressed in the blood of adult individuals administered a trivalent inactivated influenza vaccine (TIV), genes involved in type I IFN signaling, such as *STAT1*, *IF135*, *IRF7 IFIT1*, *MX1* and *IRF9*, were markedly overexpressed less than 24 h post-vaccination [19]. Furthermore, the expression patterns of type I IFN-related genes strongly correlated with the magnitude of the antibody response at 14 and 28 days after vaccination. Additionally, this correlation was stronger in high responders than in low responders to vaccine stimulation [19].

Comparison between the transcriptomic profile of a TIV-influenza vaccine and a live attenuated influenza vaccine (LAIV) in PBMCs from the blood of adult individuals showed the overexpression of several type I IFN-related genes (Table 2), among others, 3 days after LAIV but not TIV injection [17]. A similar finding has been reported by Zhu et al. (Table 2), although the induction of type I IFN-related genes was observed in the blood of a cohort of newborns 7–10 days post-vaccination [110]. In another study, the expression of a single type I IFN-related gene was induced by TIV in the blood of adult vaccinees on day 1 post-vaccination (Table 2) [111]. Also, the overexpression of seven genes that were apparently not relevant to the immune response was observed [111]. These seven genes were strongly correlated with the magnitude of the antibody response, suggesting that the induction of their expression could be crucial for TIV immunogenicity [111].

Results from the laboratory of Cao et al. disagreed with these findings relative to the TIV but agreed with the results relative to the LAIV. Cao et al. found that type I IFN-related genes, such as *OASL*, *OAS3*, *IFIT1*, *IFIT3*, *IFI44* and *ISG15*, were overexpressed after one day post-vaccination in TIV-treated children’s groups of all ages and after 7 days post-vaccination in the LAIV-treated children’s groups but especially in the younger participants [31]. Furthermore, transcriptional profiles in LAIV vaccinees, in addition to being more attenuated compared to TIV vaccinees, showed minor robustness and correlations between the expression of type I IFN-related genes and antibody titers. Other research groups, previously mentioned in this section, have demonstrated that TIV can induce an IFN-based signature in adult individuals as early as 24 h post-vaccination [19,20]. Thus, it seems that LAIV but not TIV has some dependence on the age of the vaccines.

When MF59-adjuvanted and non-MF59-adjuvanted TIVs were administered to infants enrolled in a clinical trial (Table 2), the overexpression of type I IFN-related genes such as *OAS1*, *OAS3*, *IFIT1*, *IRF7* and *IFIH1* was evident, and it was also correlated with an early antibody response, although transcriptional responses to TIV in infants were less robust and not observed in all the vaccinated infants compared to responses observed in adults [109].

Comparison between the transcriptomic profiles of LAIV and TIV influenza vaccines and a live attenuated yellow fever vaccine [16,22] in human PBMCs, either isolated from the blood of vaccinees or stimulated in vitro with these vaccines, revealed that *IRF7*, *OAS* and *MX1* were commonly induced by yellow fever and LAIV influenza vaccines and that the overexpression of type I IFN-related genes correlated with the magnitude of antibody responses to influenza and yellow fever vaccines [17]. These findings suggest that different influenza vaccines induce distinct molecular signatures in the blood, which can be common to other vaccines. Furthermore, through longitudinal transcriptional profiling of PBMCs (from five influenza seasons) from the blood of TIV-vaccinated individuals, Nakaya et al. showed that signatures of innate immunity are shared across multiple influenza seasons and in heterogeneous vaccinated populations [17].

Results by Gonçalves et al. showed that TIV immunogenicity and biomarker gene expression are influenced by the route of immunization. In fact, the transcriptomic analysis of the blood of healthy subjects vaccinated by subcutaneous (s.c.), intradermal (i.d.) and intramuscular (i.m.) administration of a TIV revealed that compared to expression before administration, 389 genes were differentially expressed in the intramuscular-treated subjects, 127 in the intradermal-treated subjects and 4 in the subcutaneous-treated subjects 24 h after vaccination. Twenty-four genes were common among these routes. Among the total differentially expressed genes (496), type I IFN-related genes such as *STAT1*, *IRF9*, *IFI35* and *IRF7* were significantly upregulated, especially by the i.d. and i.m. routes [18]. These findings collectively reveal heterogeneity in responses to the TIV.

Overall, these interesting findings clearly suggest that vaccine characteristics, route of administration and age of vaccinees strongly influence the kinetics, robustness and extent/patterns of expression of type I IFN-related genes induced by vaccines. Furthermore, the vaccine administration route and the age of vaccinees appear to play a role in the vaccine effectiveness. Also, these findings confirm the potential utility of type I IFN-related genes as valuable biomarkers for type I IFN-associated vaccines, but they also highlight the existence of other genes related to the antigen as an important determinant of vaccine immunogenicity. Additionally, it deserves to be underlined that signatures of innate immunity are shared across multiple influenza seasons and can be common to other vaccines.

### 3.3. Methods, Assays and Systems Used to Identify Biomarkers for Vaccine Safety

Investigated vaccines and adjuvants: pertussis vaccine and toxicity reference pertussis vaccine (whole-cell inactivated pertussis vaccine), inactivated monovalent A/H5N1 whole-virion influenza vaccine adjuvanted with aluminum hydroxide (PDv), inactivated whole trivalent influenza vaccine (WPv), trivalent HA influenza vaccine (HAv), inactivated influenza virus vaccine used as a toxicity reference vaccine and concentrated bulk materials of influenza HA vaccine, reference influenza vaccine, subvirion influenza vaccine (HAv), poly I:C adjuvanted hemagglutinin split vaccine HAv, AddaVax™-adjuvanted HAv (squalene-based oil-in-water emulsion adjuvant similar to MF59, which is a safe and potent adjuvant for use with human vaccines), non-adjuvanted hemagglutinin split vaccine and toxicity reference influenza vaccine (whole-virion inactivated influenza vaccine).

Vaccine safety assessment and biomarker identification were performed using lungs isolated from vaccine-treated wild animals and humanized NOG mice, as well as human PBMCs stimulated with vaccines. Gene expression and transcriptomic analyses were performed by quantitative RT-PCR analysis, DNA microarray, QuantiGene Plex (QGP) assay and hybridization with branched DNA (bDNA) amplifier combined with multi-analyte magnetic beads.

Data from gene expression microarrays and gene ontology were analyzed and profiled using two-dimensional hierarchical clustering and multiple regression analysis.

### 3.4. Correlates of Vaccine Safety

The evaluation of vaccine safety using a systems biology approach (Figure 4) was introduced for the first time by a Japanese research group, who successfully identified potential toxicity-related biomarkers for the pertussis vaccine in two independent studies (Table 3) and demonstrated that the identified biomarkers could be used to evaluate the batch-to-batch consistency and safety of vaccines [24,115].

This group has also focused part of its research on evaluating the safety of influenza vaccines using a set of molecular biomarkers that were previously identified and characterized in rat lungs using a DNA-based transcriptomic approach [25]. This biomarker set was composed of 17–20 genes (*Irf7*, *Lgals9*, *Lgalsbp3*, *Cxcl11*, *Timp1*, *Tap2*, *Psmb9*, *Psme1*, *Tapbp*, *C2*, *Csf1*, *Mx2*, *Zbp1*, *Ifrd1*, *Trafd1*, *Cxcl9*, *β2m*, *Npc1*, *Ngfr* and *Ifi47*), which were significantly and particularly overexpressed in rat lungs after vaccination with a whole-virion-particle inactivated influenza vaccine. Several genes of the biomarker set were type I IFN-induced or -inducible genes. Alterations in gene expression in the lungs have been used as a parameter to evaluate the safety of the influenza vaccine. The lungs were chosen for such an investigation because they allowed researchers to clearly distinguish differences in gene expression between a whole-virion-particle inactivated influenza vaccine (WPv: toxicity control vaccine associated with severe adverse reactions) and a hemagglutinin split influenza vaccine (HAv: safety control vaccine associated with mild adverse reactions), as compared with other organs.

Advanced computational analysis has revealed that the level of expression of these biomarker genes is well correlated with toxicological results from the conventional ATT (the abnormal toxicity test, which is based on the control of the body weight of animals) and LTT (the leukopenic toxicity test, which is based on checking leukocyte counts in the blood of animals) animal tests implemented for quality and safety control of influenza vaccines in Japan [25].

The highlighted utility of the (17–20)-biomarker set as a tool to evaluate the safety of vaccines was exploited by the same group to evaluate and compare the safety of batches of seasonal trivalent HA influenza vaccine (HAv) from four different manufacturers [26]. The results obtained from that study showed that the expression of some biomarkers of the (17–20)-biomarker set, including type I IFN-related genes such as *IRF7* and *MX2*, was significantly increased in the lungs of rats treated with a batch from one manufacturer but not in the lungs of rats treated with the control and batches from the other three manufacturers. Moreover, the expression level of the identified biomarkers correlated with the results of the conventional ATT animal test.

Additionally, based on the percentage of the relative expression level of biomarkers induced by HAv compared to that induced by the toxicity reference vaccine (WPv) used in the LTT animal test, *MX2* and *IRF7* were classified as “grade 1”, while other type I IFN-related genes such as *IFI47* and *TRAFD1* or *IFRD1* were, respectively, classified as “grade 2” or “grade 3”, being 10%, 20% and 40%, respectively, less expressed in HAv-treated lungs compared to WPv-treated lungs. This classification is based on established criteria, which determine that leukocyte loss in test samples must not be greater than 20% of that of the reference toxic vaccine, known to be capable of inducing a strong loss of peripheral leukocytes [34], or less than 50% of that of the control vaccine [26].

In that same study, furthermore, the evaluation of vaccine safety using the (17–20)-biomarker set not only allowed researchers to distinguish between the different manufacturers’ batches but also enabled them to discriminate between adjuvanted and nonadjuvanted influenza vaccines, showing higher and more accurate detection sensitivity than that achieved with animal models [26].

In light of the demonstrated greater sensitivity and accuracy of the (17–20)-biomarker gene set in evaluating vaccine-related toxicity in comparison to conventional animal testing, the same group also exploited this set to select potential biomarkers related to leukocyte reduction in the blood that can substitute the LTT for the safety evaluation of influenza vaccines [28]. Thus, performing multiple regression analysis of the biomarker genes expression profile in HAv-treated mice lungs revealed that the expression levels of seven genes of the biomarker set, among which four were type 1 IFN-related genes (*TRAFD1*, *IRF7*, *IFI47* and *IFRD1*), can predict with exceptional accuracy (>90%) the leukocyte reduction in the blood because they were significantly associated with the leukocyte reduction measured in the blood of HAv-treated mice in comparison to mice treated with saline or a reference vaccine. This inverse correlation between the gene expression level in the lungs of vaccine-treated mice and leukocyte reduction level in the blood is in line with the LTT method [28]. The involvement of type I IFN signaling in leukocytes reduction is not surprising since type 1 IFN-induced leukopenic reactions have previously been observed [34]. Ato et al. demonstrated that influenza-induced IFN-α/β is produced concomitantly with leukocyte loss in the blood of mice and is involved in leukocyte apoptosis.

The expression profile of the (17–20)-biomarker gene set has also been shown to correlate with ATT and LTT test results in diluted samples of influenza vaccine and in a dose-dependent manner [27]. Indeed, after treating rats with serial dilutions of a toxicity reference influenza vaccine, all the genes of the biomarker set showed dose-dependent expression profiles, which correlated well with the results of the ATT and LTT tests, compared to rats treated with saline solution. Furthermore, most, but not all, genes were upregulated in a dose-dependent manner in all animal groups treated with serial dilutions of influenza HA vaccine bulk materials, compared to saline-treated groups. Type I IFN-related genes *MX2*, *IRF7* and *IFI47* were upregulated, but *IFRD1* was not affected at all. On the other hand, the ATT and LTT animal tests did not consistently provide dose-dependent responses when diluted bulk materials of influenza vaccines were tested, suggesting that there are differences between these materials and that these differences are not influenced by dilution.

These findings by Momose et al. were obtained using bDNA technology (Table 3), which has been shown to have higher detection sensitivity than conventional animal safety tests used for influenza vaccines. In fact, bDNA technology can detect and quantitate multiple mRNA targets simultaneously in a single step and in less time than conventional safety testing. Altogether, this suggests that, in contrast to gene expression analysis, animal safety tests can obviously indicate whether the tested vaccine is diluted or not when animals are treated with reference vaccines, but not with bulk materials of influenza vaccines [27]. This, in turn, suggests that analysis of gene expression using an ex vivo vaccine-treated tissue, such as rat lung, is more precise and perhaps even more reliable than conventional animal testing.

The (17–20)-biomarker gene set was also useful in determining the impact of the route of administration on the vaccine safety profile [30]. That investigation was made possible through the development and validation of mathematical regression equations that consider several parameters, such as body weight, leukocyte count and the expression level of lung biomarker genes, which were all determined in mice inoculated with reference or control vaccines or adjuvants for toxicity via i.m., intraperitoneal (i.p.) and intranasal (i.n.) routes (Table 3). Toxicological changes in body weight and leukocyte count were determined according to the ATT and LTT animal toxicity tests, which are relevant to evaluate the safety of the vaccine administration route. Then, the probability that the set of biomarker genes would be classified as having similar immunotoxicity and immunogenicity to the reference vaccine was calculated. Furthermore, safety categories have been established that classify each injected material according to its level of toxicity, such as category I (with non-RE toxicity) for the saline and non-adjuvanted HAv vaccine, category II (with almost RE-like toxicity) for the vaccine-adjuvant toxicity control poly I:C and HAv vaccine combined and category III (with RE-like toxicity) for the reference toxicity vaccine (Table 3).

In that study, the developed regression equations allowed researchers to associate the toxic potential of the vaccine inoculation route with the established expression level of the lung (17–20)-biomarker gene set. In addition, they led to the selection of 12 genes, including *MX2* and *IFI47*; 10 genes, including *TRAFD1*, *IRF7*, *MX2* and *IFI47*; and 5 genes, including *IRF7* and *IFI47*, as biomarkers to predict the safety of the i.n., i.m. and i.p. routes, respectively [30].

Thus, the immunotoxicity of vaccines and adjuvants related to each administration route was inferred from the expression profiles of the set of biomarker genes in the lungs of vaccine-treated mice.

Worth noting, in this approach proposed by Sasaki et al., despite having no obvious immune functions, the lung is confirmed as a key organ that intervenes promptly in response to the administration of the influenza vaccine regardless of the route of administration [30].

This research group has previously investigated the impact of the administration of the influenza vaccine by the i.n. route on the biomarker gene expression profile in the lung, obtaining results that support the profile described above [29]. In that previous study, after i.n. administration of an adjuvanted toxicity reference influenza vaccine to mice, a dose-dependent expression of the (17–20)-biomarker gene set was observed in mouse lungs. The expression of biomarker genes correlated with typical vaccine- or adjuvant-induced morphological changes in the lung and nasal mucosa, as demonstrated by histopathological analysis [29].

The (17–20)-biomarker gene set recently (partially) predicted influenza infection-related leukopenic toxicity and inflammatory reactions in a pDC-retaining humanized mouse model obtained by the intravenous engraftment of human PBMCs in NOG (NOD.Cg-prkdcscidil2rgtm1Sug/Jic) mice (referred to as the short-term model (ST model) and involves pDCs and CD 14+/SSC^high^ cells in the lungs) [116]. Indeed, following the administration of a reference vaccine having well-known leukocyte-related toxicity, only an increase in the expression levels of marker genes was detectable in the ST mice lungs but not the expected reduction in leukocyte count, which it became detectable later, as compared with the control group. This result may suggest that changes in biomarker gene expression occur before leukoreduction, and this confirms the utility of marker genes as reliable predictors for vaccine immunotoxicity, at least in the ST model.

Among the marker genes induced in the lungs of ST mice, the type I IFN-related gene *MX2* was significantly upregulated [116]. This finding suggests that some adverse effects of the toxicity reference vaccine are mediated by type 1 IFN-related signaling. Accordingly, previous studies have reported the induction of *IFI47* and *MX2* by type 1 IFNs after infection with whole-virion inactivated influenza virus [55]. Such toxicity of type I IFNs is likely exerted through pDC-mediated reactions since pDCs are major producers of type I IFNs and are required for whole-virion inactivated influenza vaccine leukocyte-related toxicity in mice [34].

Because PBMCs contain immune cells, such as monocytes, DCs and lymphocytes (T-, B- and NK cells), which are crucial for the immune function, have an in vitro immunological profile that reflects their in vivo profile [118] and can express the (17–20)-biomarker gene set [116] originally identified in animal lungs, as described above, PBMCs were used to further evaluate and confirm the utility of the (17–20)-biomarker gene set as a tool for the safety assessment of influenza vaccine in humans.

After stimulating PBMCs isolated from healthy adult donors with HAv influenza vaccine or toxicity reference influenza vaccine in vitro (Table 3), five genes of the (17–20)-biomarker set showed a significant alteration in their expression level [117]. Among these five genes, the type I IFN-related genes *MX2*, *IRF7* and *TRAFD1* were significantly overexpressed at all tested doses compared to the control group but not in a dose-dependent manner. Furthermore, the expression levels observed with the HAv vaccine were lower and differed between donors compared to the expression levels observed with the reference vaccine. In addition, these biomarkers were found to be particularly overexpressed in DC subtypes present in PBMCs. This finding is not surprising since DCs subtypes are known to be one of the first immune cells that readily react to vaccine/adjuvant administration.

In summary, this robust evidence demonstrates that the proposed powerful transcriptomic approach is indeed capable of assessing the safety of different types of influenza vaccines and adjuvants by measuring alterations in the expression profile of a set of biomarker genes. These biomarker genes have been shown to be subject to expression alteration by influenza vaccines in wild type and humanized rodent lungs in vivo/ex vivo and in human PBMCs in vitro. The expression level of these biomarkers could be influenced by the vaccine’s composition or other variables. Their expression levels have been shown to be significantly correlated with indicators of immunotoxicity (body weight loss and leukopenic toxicity) obtained from conventional animal tests. Also, they have been shown to be useful for the safety evaluation of influenza vaccines regardless of the administration route and dose, the manufacturer and whether they are adjuvanted or nonadjuvanted. Moreover, these biomarker genes showed higher sensitivity, robustness and accuracy compared to conventional animal tests.

## 4. Potential Limitations That May Hamper Vaccine Characterization Using Alternative Non-Animal Models

As detailed above in the introduction, non-animal methods alternative to the bioassay are urgently needed for vaccine assessment due to the numerous limitations inherent to in vivo testing and also due to the need to apply the 3Rs principles. Regardless of the substantial advantages of using non-animal systems and the encouraging and robust proof-of-concept evidence on the availability of potential molecular biomarkers for the evaluation of vaccine safety and efficacy, non-animal systems present several limitations that must be overcome before implementing them for vaccine assessment. Potential limitations that may hinder the characterization of vaccines in in vitro methods are related to the immune system, the technical procedure, the experimental design, the vaccines and their composition, the vaccinated donors, the pre-exposure to the antigen, the individual immunogenicity/reactogenicity (low responders and high responders), age, gender, ethnicity and others. Each of these limiting factors can potentially interfere with the accuracy, precision, reliability, robustness and homogeneity of biomarkers identified in vitro.

A major concern is the complexity of the immune system itself and its reaction in response to the vaccination. This is challenging because the immune response is very complex, being composed of multiple responses, whereas in vitro methods detect the single profile of the immune response [19,119]. This substantial difference could produce a significant variation in the extent of the measured immune signatures. The immune signature could also be substantially influenced by the heterogeneity of the human innate immune system [108]. This heterogeneity is caused by an individual variability in the development of some innate cells, such as DCs.

The huge amount of data generated by advanced high-throughput techniques and related technical aspects could influence the reliability of the result. Indeed, data acquisition, management and modeling, as well as a high level of noisy background, can increase the probability of artifacts and contradictions [120]. For example, differences in the expression values of some genes measured by two different detection methods, such as RT-PCR and branched DNA (bDNA) used in transcriptomic analysis, have been described [27]. This discrepancy between the two methods could be due to differences in target RNA extraction efficiency and/or the fact that the bDNA assay identifies and quantitates multiple mRNA targets simultaneously in a single step and within the detection sensitivity of the instrument, while the assay showed a different sensitivity for genes with different expression [27]. Furthermore, critical limitations could arise from the experimental design itself and could concern the effect of age and sex on vaccination outcome, the generation of non-specific effects, the type and size of the sample, the timing of sampling and the data analysis [31,121].

Bucasas et al. have found that the pre-exposure of vaccinated donors to the tested antigen through an infection or through a previous vaccination impedes a correct calculation of the antibody titer and consequently impedes a correct correlation between the antibody response and the expression level of the marker gene [19]. This limitation deserves particular attention when studying correlates of the immune response to the vaccine that could be used in the future for the development of alternative models.

PBMCs have several advantages, as mentioned above [117]. However, because they are composed of various cell populations, only those distinct populations could differentially express genes [122]. Additionally, PBMCs may not be suitable for the detection of recall responses because their content in memory cells is not sufficient for the onset of such a response [122]. Similarly, gene upregulation in cells that are not predominantly present in PBMCs may not be captured. Furthermore, some expressed genes and proteins may be subject to alteration due to the nonphysiological growth, expansion and differentiation of cells positioned in two dimensional (2D) monolayers [123,124]. Therefore, 2D cell cultures can partially reflect the in vivo environment present in immune cells and tissues. This may cause differences in the quality and quantity of biomarkers captured in vitro compared to those of in vivo biomarkers.

Age, sex, geographical origin and ethnicity are among the most important potential limitations for the determination of vaccine correlates using in vitro models. These are the variables that can most influence the immune response to the vaccine and, consequently, vaccine correlates due to the significant differences in the responsiveness of heterogeneous groups. For this reason, Nakaya and Pulendran proposed to preliminarily characterize the immune response and associated molecular correlates produced in these groups by vaccination [119].

Bucasas et al. [19] found that type I IFN-related genes were more overexpressed and strongly correlated with the magnitude of the antibody response in high responders to the influenza vaccine compared to low responders, who showed a profile poorly related to the immune response, as mentioned above (see Section 3.2). This heterogeneous response to the same vaccine represents a major challenge in identifying robust biomarkers that can be used for the development of non-animal models. It has been reported that an alteration in the methylation of transcription factors and genes that play a role in antigen presentation may be responsible for the poor responsiveness to vaccination [125,126].

Finally, the lack of standardized assays is also one of the most important limitations. Thus, the standardization of assays across different laboratories may overcome or avoid many of the illustrated issues.

## 5. Discussion

The transcriptomic approach, using in vitro and ex vivo models or blood samples from vaccinated human subjects, allows researchers to predict vaccine efficacy and safety. This is possible through the identification of vaccine-associated immune signatures whose expression level is significantly correlated with the magnitude of vaccine immunogenicity (vaccine-related neutralizing antibody and antigen-specific CD8+ T-cell responses) and/or reactogenicity (vaccine-related body weight loss and reduction in blood leukocytes) determined by in vivo measurements. Considering that vaccines with well characterized efficacy or toxicity profiles were used in the studies analyzed in this article, the expression levels of the identified biomarkers could be considered to correlate with the efficacy or toxicity of those vaccines as well.

This approach also revealed that vaccine-associated immune signatures for both efficacy and safety predominantly belong to the type I IFN signaling pathway, indicating the wide spectrum of involvement of type 1 IFN regulation in the immune response to vaccines. Indeed, type I IFNs are significantly involved in the process of differentiation of B cells into antibody-secreting plasma cells [114] and in the leukocyte apoptosis and leukopenic toxicity induced by severe influenza infection [34]. Type I IFN signaling becomes toxic against host cells in the presence of chronic infections, which implies its sustained production over time and subsequent onset of IFN-mediated pro-inflammatory cytokine storm and oxidative stress [127]. In such a situation, ISGs such as *IRF7*, *MX1* and *ISG15* are unable to elicit antiviral effects against some severe influenza strains resistant to these genes [128,129,130], while *IFTIM* can elicit antiviral effects and circumvent fatal events associated with severe influenza pandemic viruses [131]. The failure of *IRF7* and *MX* genes in protecting the host is due to their insufficient endogenous amounts, which are unable to control virus replication in the presence of severe and virulent influenza virus [128,129,130] thus facilitating virus replication and subsequent deleterious effects on the host [132]. While the antiviral protective mechanism of *IRF7* in the immune response is well defined, there is evidence suggesting that *MX1* may exert its anti-influenza protective effect by downregulating factors such as IL-6, IL-1β and TNFα, among others, involved in excessive cytokine response [132]. This mechanism of action, in addition to its nuclear localization, which confers it a powerful antiviral restriction action, and the fact that it exerts its effects in a very early stage of the replication cycle [133], could together be responsible for the important role of *MX* in the immune response to the vaccine.

In this investigational study, key ISGs such as *IRF7*, *MX1*, *MX2* and *OAS1* have been revealed to be key biomarkers for both vaccine efficacy and safety, as shown in Table 2 and Table 3. This suggests that an absolute classification of vaccine-induced immune signatures as toxicity- or efficacy-related biomarkers may not be correct because the same biomarkers could be involved in both vaccine outcomes. For this reason, in the studies overviewed in this article, the expression levels of biomarker genes induced by a reference vaccine (for toxicity or efficacy) were used as judgment criteria according to which the same biomarkers induced by a test vaccine were judged to be toxic, effective or both based on whether the values of their expression are higher or lower than those of the reference vaccine. Thus, vaccines with a reactogenicity higher than that of the reference vaccine are considered unsafe regardless of their high efficacy because this high reactogenicity predicts that the test vaccine would have a potential risk of developing toxicity higher than the potential risk that the reference vaccine has [117].

This method is not the only one suggested to discriminate between type I IFN-related biomarkers for vaccine efficacy and safety. Chan et al. showed that the expression of molecular correlates of immunogenicity, including type I IFNs, occurs at timepoints of the immune response that differ from the timepoints at which the expression of molecular correlates associated with systemic AEs occurs in the context of the live-attenuated yellow fever vaccine administered to adults. These authors also demonstrated that the levels of neutralizing antibody titers against the yellow fever virus were similar both in the presence and absence of AEs, suggesting that reactogenicity can coexist with immunogenicity without impairing it [32]. Adverse effects such as fever, myalgia and headache, occurring mainly at day 1 after vaccination for yellow fever, were not toxic but caused by a proinflammatory process that is part of the innate immune response itself and is usually associated with strong vaccines [134,135]. For this reason, the treatment of AE symptoms during this specific window of time is not recommended because such treatment would also impair the later adaptive immune response [136].

Importantly, the data analyzed in this paper were obtained using human-derived models, such as human cell cultures stimulated with different types of vaccines and/or blood samples from vaccinated human individuals of different ages, demonstrating that the transcriptomic approach could provide not only reliable biomarkers to study, predict and monitor the outcome of vaccines but also an alternative method to the animal model. This approach may replace the use of animals directly without the need for any extrapolation, adaptation or transformation for their application to the vaccine efficacy assessment once both the reliability and robustness of the identified biomarkers and their equivalent performance to the animal testing have been demonstrated [137,138]. Furthermore, the transcriptomic approach would lead to a substantial reduction in the number of animals used in evaluating the safety of vaccines. In comparison to the ATT and LTT conventional animal tests, which require a large number of animals to fulfill the requirements of a valid assay and obtain reliable results [139,140], the transcriptomic approach requires a small number of animals necessary for vaccine administration and the development of the immune response, which is then analyzed to identify safety biomarkers in an ex vivo system using lungs isolated from vaccinated animals. Thus, lungs tissue can provide multiple measurements from one vaccinated animal, and this would replace animals that would have been used as replicates to obtain a correct evaluation and sufficient information from the ATT and LTT animal tests. Additionally, the transcriptomic approach would spare the animals from unnecessary pain and suffering, since the animals will be euthanized and the lungs isolated for subsequent analysis before the development of severe immune reactions. Furthermore, the use of human cell cultures stimulated with vaccines might in the future lead to the replacement of conventional animal safety tests after adequate validation of the identified potential biomarkers.

Although the overall results hold promise for a potential assessment of vaccine efficacy and safety in vitro, the following aspects remain critical:(1)Due to the dual effect of type I IFNs on the immune response, the reliability of the identified biomarkers in their ability to distinguish between vaccine efficacy and safety outcomes could be questionable.

As mentioned above, type I IFN effects can be beneficial and protective when their production is limited, such as during acute infections, or they can be deleterious, causing autoinflammation or autoimmune disorders when their production is sustained overtime, such as during chronic infections [141,142]. Therefore, type I IFN production should be maintained under control to achieve optimal levels capable of conferring good protective adaptive immunity without causing major adverse effects [142]. To this end, strategies have been developed to reduce or prevent the prolonged production of type I IFNs [143] that occur after vaccination with certain types of vaccines or modify the affinity and availability of type I IFNs for IFNAR after vaccination [144,145,146]. For example, Palacio et al., using different viruses, routes, challenges and host genetic backgrounds, showed that a brief interruption of type I IFN signaling during acute infection can improve the adaptive immune response and immunological memory, resulting in protection against reinfections [147].

(2)Considering that type I IFN cellular sources, magnitude and timing of production, signaling and production of ISGs can differ in the host as well as in different infection models based on the infectious load and the route, type and stage of infection or vaccination, as well as host genetics and the immune-regulatory environmental background, type I IFN-related biomarkers associated with vaccine efficacy and safety may change as well.

It has been reported that the route of infection/vaccination can substantially affect the production of type I IFNs and their regulatory and immunomodulatory effects, thus influencing the disease/vaccine outcome. Accordingly, different research groups have demonstrated that type I IFNs can negatively impact the efficacy of mRNA-based vaccines and induce the development of AEs when vaccines are administered by s.c., i.d., intranodal and i.m. immunization routes but not the intravenous (i.v.) route, although efficacy and immunogenicity remain comparable [148,149,150,151]. The influence of the administration route on the type I IFN effects was also observed among the non-i.v. routes. A recently published systematic review revealed that the i.m. route is more immunogenic and associated with fewer systemic AEs than the s.c. route for almost all vaccine types (adjuvanted and non-adjuvanted live virus or inactivated whole cell, split cell and subunit) [152]. In contrast, Syenina et al. found that immunization with an mRNA-based vaccine by the s.c. route causes fewer AEs than the i.m. route [153]. This contradiction could be due to differences in the amounts of type I IFN induced by the two routes that may in turn be due to differences in the vaccine’s type and formulation, doses administered or experimental conditions. In those studies, the antigens that made up the tested vaccines were different. It has been demonstrated that the type of antigen/virus can determine the type of cell that produces type I IFNs in certain tissues. For example, in mouse lung tissue, type I IFNs were produced by epithelial cells after i.n. infection with the influenza A virus [154] and by alveolar macrophages after i.v. infection with Newcastle disease virus (NDV) [45]. Regardless of this discrepancy, mechanistically, type I IFNs can promote T-cell proliferation, survival and differentiation into cytolytic effector cells when the exposure of type I IFNs occurs after T-cell receptor (TCR) activation by DCs, as they can also inhibit proliferation and induce the apoptosis of T cells when type I IFN exposure occurs before T-cell receptor activation by DCs [149,151,155]. In the case of the s.c. route of immunization, type I IFNs bind to the IFNAR present on T cells before being activated by DCs, thus eliciting an inhibitory effect on the protective immune response through the exhaustion of T cells. In contrast, in the case of the i.v. route of immunization, spleen DCs reach resident T cells and activate them before type I IFNs. This right timing of T-cell activation via the i.v. route is facilitated because mRNA-based vaccines are delivered not only to DCs but also to macrophages to produce a strong T-cell-mediated immune response. Furthermore, the systemic targeting of DCs by type I IFNs allows for DC maturation and the development of antigen-specific T-cell responses [156]. Additionally, the expression of USP18 in macrophages of some certain immune cells aids in limiting the effects of type I IFNs [149,151].

Together, these findings show that the immunization route can substantially influence the timing of action, the type of producing cell and the produced amount of type I IFNs, as well as the order of TCR activation, thus determining whether type I IFN signaling is protective or detrimental. As consequence, these variables could cause significant variation in the expression levels of type I IFN-related biomarkers, thus affecting their utility as indicators of efficacy or toxicity. For example, low amounts of type I IFNs stimulate the optimal production by DCs of L-12p70, also termed IL12, and play an important role in the regulation of cell-mediated immunity [157], while high quantities of type I IFNs inhibit the production of L-12p70 by human monocytes/macrophages with consequent effects on the immune response [158]. It has been reported that the amount of the antigen load and the duration of exposure can significantly influence type I IFN production and related signaling. Indeed, in the presence of a very low antigen load, the expression of type I IFNs and the induction of ISGs are low and, consequently, the onset of T-cell-mediated protective immunity could be indirectly hampered. In fact, in such a condition, the innate antiviral action of type I IFNs induces an early and prompt reduction in the antigen load to levels insufficient to activate T cells and efficiently elicit immune responses [155,159]. Instead, a high antigen load can lead to a chronic infection characterized by increased type I IFN-induced expression of ISGs and inhibitory molecules, such as programmed cell death protein 1 (PD-1) and interleukin 10 (IL-10), which are involved in the T-cell proliferation and survival state [160]. In an attempt to clear substantial infection, the persistent stimulation of T cells causes their exhaustion and, consequently, dampens the adaptive protective immune response, facilitating viral persistence [160]. Using IFNAR-blocking antibodies [144,145,146] or targeting negative regulators of type I IFNs, including OASL1, USP18 and ISG-15 [161,162,163], has been proposed as a strategy against the viral persistence phenomenon. Instead, inducing type I IFN signaling and ISGs, such as *MX1*, *IRF7* and *USP18*, as well as CD8+ T-cell response and the transient depletion of regulatory T cells (Tregs) by calcium phosphate (CaP) nanoparticles (NPs) functionalized with CpG ligand and viral peptides, has been proposed as a strategy to reduce viral loads in chronic retroviral infections because in such a condition, type I IFNs induce the reactivation of CD8+ T-cell responses [164,165,166].

(3)Whereas the identified biomarkers may be applicable to vaccines belonging to the same vaccine type analyzed in this paper, whether these same biomarkers would be effective in predicting the immunogenicity and toxicity of vaccines belonging to other vaccine types, such as mRNA-based, vector-based or subunit-based vaccines, among others, remains unknown. The literature provides many examples of vaccines in which the production of type I IFNs is negligible or is exaggerated without benefit.

For example, as mentioned above, mRNA-based vaccines rely on the production of type I IFNs, which negatively affect their efficacy. This negative effect was initially attributed to the inhibition of the expression of antigen-encoding mRNA protein by type I IFNs [148], but it was later revealed that this negative effect and the onset of severe AEs are due to a direct and prompt action of elevated quantities of type I IFNs that occurs at the level of T cells rather than DCs [151]. Syenina and colleagues have recently shown that genes such as *T-cell immunoglobulin* and *ITIM domain (TIGIT)*, *killer cell lectin-like receptor F1 (KLRF1)* and *KLRD1*, which are involved in the exhaustion and suppression of T and NK cell cytotoxicity, show a high propensity to develop severe systemic AEs within mRNA vaccines [153]. They also showed that the overexpression of the levels of these genes is associated with the stimulation of key components of the immune system, such as monocytes and neutrophils, and the activation of genes such as IFNGR2, TLR8, TLR4, TLR1 and CSF2RB involved in immune response and inflammatory signaling as well as in the development of AEs typically associated with the live-attenuated yellow fever virus vaccination [32]. Using modified RNA nucleotides capable of controlling the release of type I IFNs is a possibility that may overcome this issue and improve the efficacy of RNA vaccines [143].

Other vaccines, such as the recombinant modified vaccinia Ankara (MVA) vaccine (a vector-based vaccine), have been shown to rely on the expression and production of high level of type I IFNs via the STING pathway, while the human adenovirus vector “Ad5” was shown not to rely on type I IFNs after injection [167,168]. The MVA-HIV vaccine exerts a stimulating effect on humoral immunity by increasing antibody levels and the activity of B cells, T follicular helper cells (Tfh) (involved in B-cell proliferation and differentiation into plasma cells and memory B cells), CD4 T cells and memory-like B cells.

Finally, chitosan, which is a promising nonviral delivery vector recently introduced as an adjuvant for vaccines, relies on type I IFN signaling and induced ISGs. It promotes cellular immunity through the STING pathway and can induce IgG2 production and Th1 responses [169]. On the other hand, a T-cell-inducing adjuvant for subunit-based vaccines has been shown to rely on the cytokines IL-12 or IL-27 rather than type I IFNs to mediate T-cell differentiation and memory formation [170].

## 6. Conclusions and Perspectives

The development of appropriate non-animal methods to evaluate batch-to-batch consistency and vaccine safety and efficacy as an alternative to animal testing might be feasible after overcoming potential limitations and taking into account some considerations, as well as performing an accurate and a careful validation and cross validation of the potential biomarkers. The demonstration within this review that the transcriptomic approach can lead to the reduction, replacement or refinement of laboratory animal use during the assessment of vaccine outcome may qualify this approach as a potential alternative to animal testing.

The fact that IRF7 and MX2 are common to different vaccines and immune responses associated with both efficacy and safety underlines their key role in the immune response against viral infections and provides further evidence of the crucial role of type I IFNs in immune defense. This finding also highlights and emphasizes the potential utility of both IRF7 and MX2 as vaccine biomarkers, at least for those analyzed in this paper. It may be possible to use the same biomarkers for both efficacy and safety evaluation, firstly because the evaluation parameters, described in Section 3.2 and Section 3.4, are based on differences between test vaccines and a reference vaccine; secondly because the context in which the vaccine’s efficacy would be assessed differs from that in which the vaccine would be assessed for safety; and thirdly because of the differences that exist between efficacy and safety biomarkers regarding the timing of production of type I IFN, its magnitude, the type of cell or tissue that produces type I IFNs and the cell type subject to type I IFN effects [32,171].

However, this optimistic conclusion may be true for the vaccines evaluated in this review but not for any type of vaccine, as discussed in detail above. Instead, this approach can be considered the first step that might pave the way towards a universal approach after the extensive characterization of the immune response and optimization of vaccine design with respect to the route of administration, antigen structure and load, as well as considering differences in age, gender and ethnicity.

While the data available in the literature suggest that potential universal predictors of efficacy and safety for any vaccine are not yet possible, in the meantime, many suggestions and proposals are reported on how to optimize the design of vaccines to improve their efficacy, reduce their toxicity and provide predictable universal molecular biomarkers. For example, Hagan’s laboratory, after analyzing 13 vaccines, including live viruses, recombinant viral vectors, inactivated viruses and glycoconjugate vaccines, proposed the kinetics of the vaccine immune response as a universal biomarker of vaccine outcome. The authors highlighted and emphasized that distinct kinetics of the immune response are shared across vaccines, although they occur at different timepoints; have peak expression of a gene module for plasma cells and immunoglobulins; and can substantially predict the magnitude of antibody responses across vaccines [172]. A previous study conducted by the same research group using a transcriptomic approach revealed and highlighted the existence of a molecular network that includes gene modules for plasma cells, immunoglobulin and B cells as predictors of the adaptive response to vaccination [21].

Some authors have proposed the use of a conserved region of the influenza vaccine antigen peptide to develop a universal influenza vaccine [173].

Others have proposed the postvaccination immune signatures and signaling pathways of innate immunity that have been defined for the yellow fever and influenza vaccines described in this paper to be used for the optimization of vaccine design [108,174].

According to others, understanding prevaccination immune-regulatory environment and the availability of prevaccination immune indicators on responsiveness to the vaccine, particularly for certain categories of the population, is important to optimize vaccine design and identify molecular biomarkers to predict vaccine outcomes [108,174].

Finally, three-dimensional (3D) cell cultures, human organoids (organ-like) and organ-on-a-chip are in vitro models that can faithfully recapitulate the structure, morphology and functions of the in vivo-originating tissues and organs [8,9]. These models consist of different co-cultured cell types, stem cells or stem cell-containing tissue explants or microscale models of dynamic tissue interfaces presented on miniaturized platforms, respectively. Considering that these models can incorporate the entire repertoire of immune cells [8], their use singularly or multiply for vaccine assessment could greatly increase the possibility of identifying biomarkers predictive of vaccine efficacy and safety that are as close as possible to in vivo biomarkers. The literature reports numerous examples that demonstrate the high efficiency of 3D systems in drug discovery approaches and in the characterization of immune responses. For example, the lung-on-a-chip system from human alveolar epithelial cells and pulmonary microvascular endothelial cells developed by Huh et al. [175] showed a pathophysiological performance and ability to assess drug efficacy and toxicity superior to those observed in 2D cultures [176]. Furthermore, inflammation-on-a-chip systems have enabled the study of the immune system, particularly regarding acute and chronic inflammation in conditions of infection and cancer (reviewed in [177]). Furthermore, different types of 3D models have enabled the assessment of the efficacy and safety of vaccines and antiviral drugs and the study of immune responses to vaccines or adjuvant candidates [178,179,180]. Finally, organoids from human lung stem cells have been shown to be able to replace ex vivo cultures of human bronchial explants in studies with influenza viruses [181].

In light of this encouraging and promising evidence, 3D technology might allow researchers to overcome most of the limitations and challenges listed above, thus shortening the distance to the ambitious idea of setting universal biomarkers for the assessment of vaccine outcomes in vitro and facilitating a reduction in or replacement of animal use, in particular for safety assessment.

## Figures and Tables

**Figure 1 vaccines-12-00583-f001:**
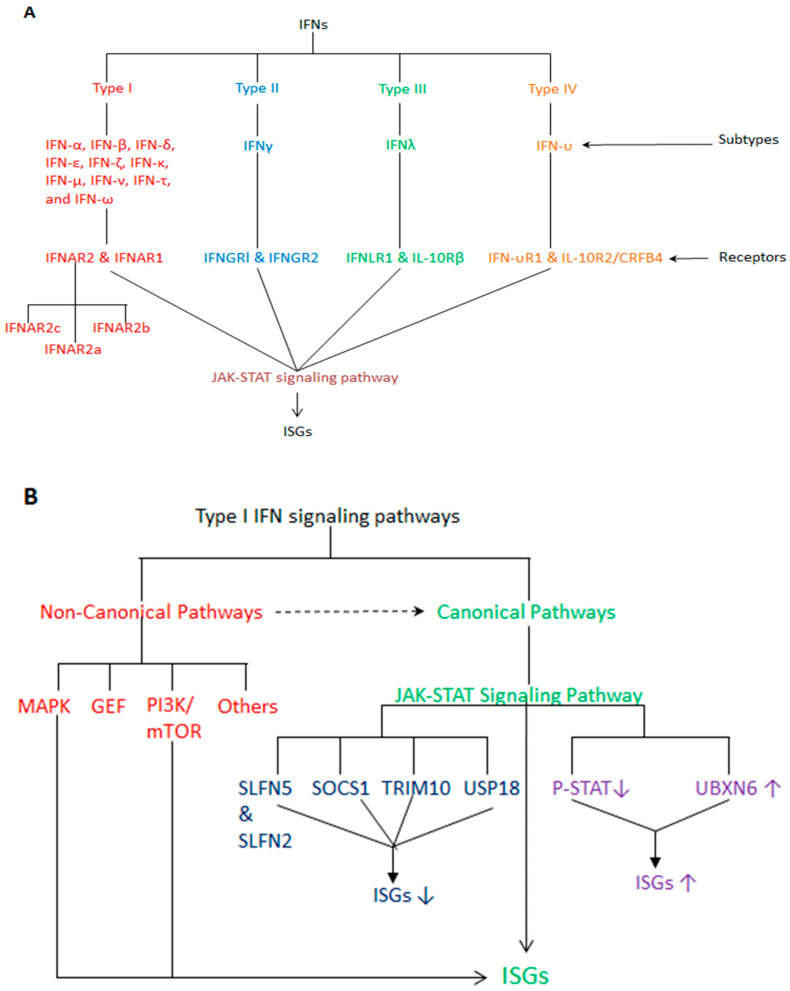
Schematic design of all existing types of interferon (IFN) and their subtypes and receptors (**A**) and signaling pathways (**B**). Figure 1A shows that all the identified IFNs lead to the production of interferon-stimulated genes (ISGs) through the activation of Janus kinase/signal transducer and activator of transcription (JAK-STAT) signaling. Figure 1B shows the main canonical and non-canonical type I IFN pathways and highlights that the non-canonical pathway can induce the expression of ISGs either directly (solid arrow) or through an interaction with the canonical pathway (dashed arrow). Figure 1B also shows the effect of stimulators (by positive feedback mechanism or direct binding to the receptor) and inhibitors (by direct binding to the receptor) of JAK-STAT signaling on the normal production of ISGs. MAPK: mitogen-activated protein kinase. PI3K/mTOR: phosphatidylinositol 3-kinase/mammalian target of rapamycin. GEF: guanine exchange factor. SLFN: Schlafen. P-STAT: phosphorylated STAT. UBXN6: ubiquitination regulatory X domain-containing protein 6. SOCS1: suppressor of cytokine signaling 1. TRIM-10: tripartite motif-containing-10. USP18: ubiquitin-specific peptidase 18.

**Figure 2 vaccines-12-00583-f002:**
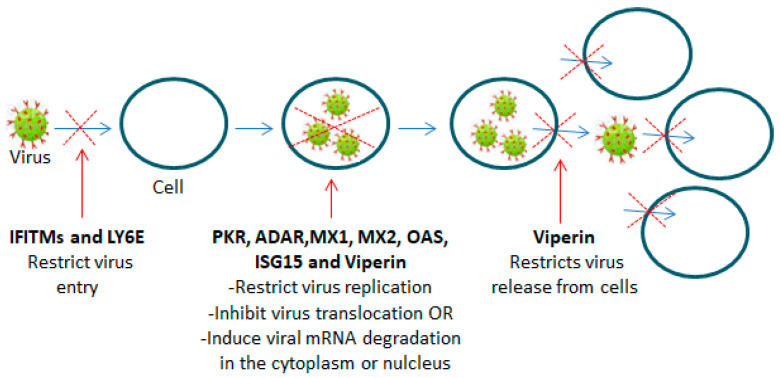
Interferon-stimulated gene (ISG) antiviral activity at the cellular level. Different ISGs act on different phases of the virus life cycle to restrict the virus spread to other cells and prevent the onset of an infection. The restriction activity explicated by ISGs and sites of action are indicated with a cross and a red arrow. PKR: protein kinase R. OAS: 2′–5′oligoadenilate synthetase. IFITM: IFN-inducible transmembrane proteins. ADAR: adenosine deaminase acting on RNA 1. MX: myxovirus resistance protein. ISG15: interferon-stimulated gene 15. LY6E: lymphocyte antigen 6 complex, locus E. Viperin: virus-inhibitory protein, endoplasmic reticulum-associated, IFN-inducible.

**Figure 3 vaccines-12-00583-f003:**
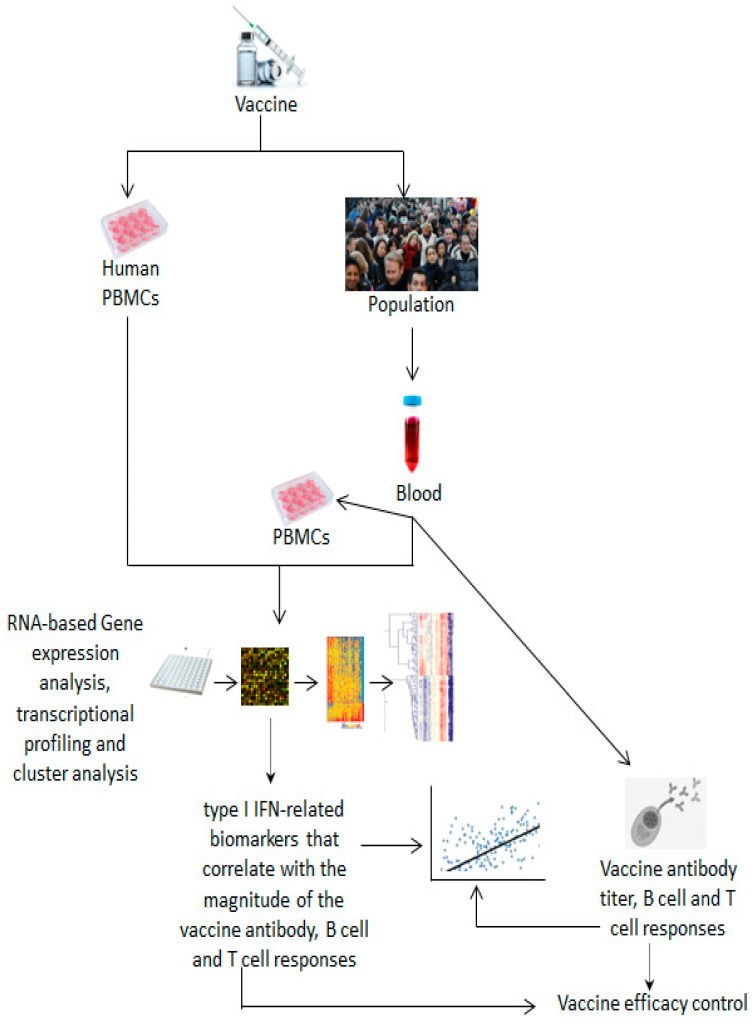
Schematic illustration of the biological approach that led to the identification of biomarkers for the evaluation of vaccine efficacy in non-animal methods. Test vaccines are added to human PBMC cultures or administered to human individuals; then, through a sequential transcriptomic analysis of PBMCs either stimulated with vaccines or isolated from the blood of vaccinated individuals, the vaccine-related immune molecules whose expression level is significantly correlated with the magnitude of the antibody response (determined in the whole blood of vaccinated individuals) were identified as potential biomarkers of vaccine efficacy. Several identified biomarkers are related to type I interferons (IFNs). The figure shows that vaccine efficacy can be assessed by using human-derived in vitro models, thus suggesting that this approach might potentially serve as an alternative to the use of animals with a full replacement. PBMCs: peripheral blood mononuclear cells.

**Figure 4 vaccines-12-00583-f004:**
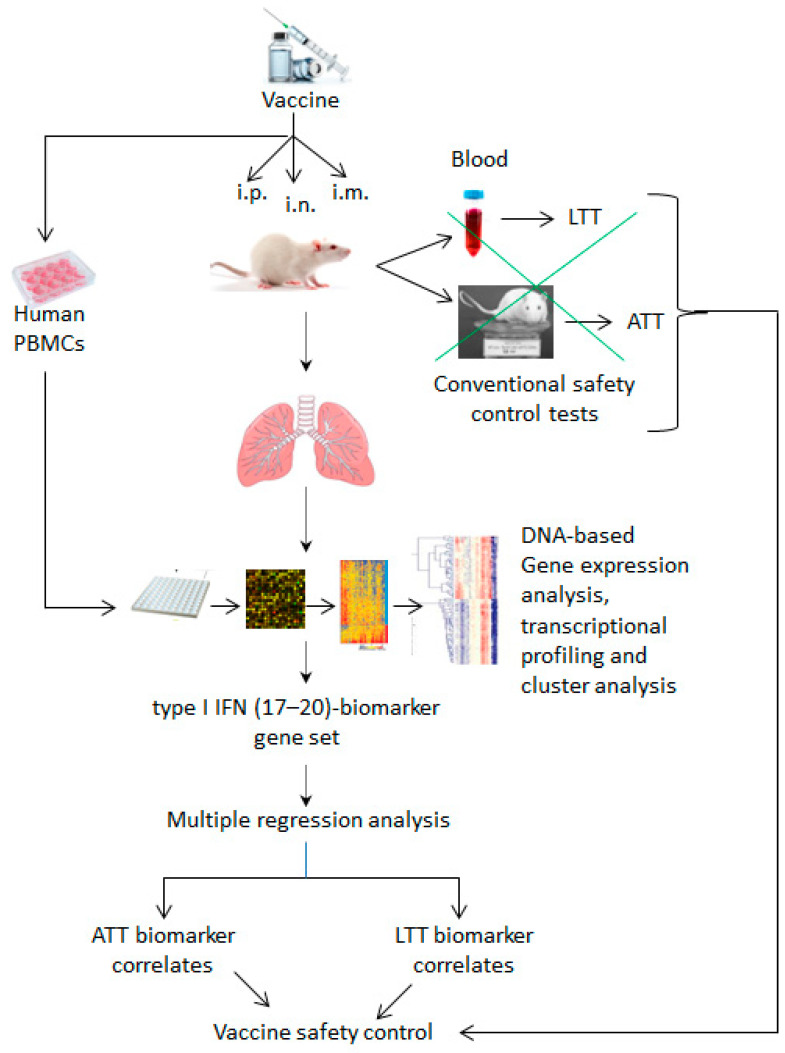
Schematic illustration of the biological approach that led to the identification of biomarkers for the evaluation of vaccine safety via non-animal methods. Test vaccines are added to a human PBMC culture or administered to animals (rats or mice); then, through a sequential transcriptomic analysis of the PBMCs stimulated with vaccines or lungs isolated from animals treated with vaccines, the vaccine toxicity-related molecules with expression levels correlated with indicators of vaccine-related toxicity, represented by a loss in body weight and leukocyte count obtained from the ATT and LTT conventional animal tests implemented for the safety and quality control of influenza vaccines in Japan, were identified as potential biomarkers for vaccine safety evaluation. ATT is based on checking body weight, while LTT is based on blood leukocyte counts. Several identified safety biomarkers are related to type I interferons (IFNs). The cross indicates that the ATT and LTT, the in vivo control tests, are going to be replaced by in vitro assays after adequate validation and the demonstration of equivalent performances to animal testing of the identified biomarkers. The figure shows that vaccine safety can be assessed by using in vitro human-derived models and ex vivo animal models, thus suggesting that this approach might potentially be able to serve as an alternative to animal testing in terms of a partial (reduction) or a complete replacement of the use of animals. i.p.: intraperitoneal injection. i.m.: intramuscular injection. i.n.: intranasal injection. ATT: abnormal toxicity test. LTT: leukopenic toxicity test. PBMCs: peripheral blood mononuclear cells.

**Table 1 vaccines-12-00583-t001:** Comparison between animal and non-animal models with respect to their convenience, practicality and efficiency in assessing vaccine outcomes.

Animal Models	Non-Animal Models
Time-consuming, expensive and laborious	Rapid, cheap and easy in terms of vaccine testing and production time and cost
Issues of precision, accuracy and sensitivity	Accurate, precise and sensitive
Poorly predictive of human responses	Highly predictive of responses, selective and specific for the target
High individual variability and low translatability to humans	Provide reproducible and robust results that can be readily applicable to humans
Issues related to supplying and maintaining animals	Practice for routine use and flexible
Not sufficiently validated	Methods are sufficiently validated and cross-validated
Procedures and treatments can cause animal suffering	The used procedures can reduce or replace animal use
Present genetic and environmental differences with humans as well as limitations in reflecting some severe adverse effects (AEs) caused by cytokine-release syndrome	-

**Table 2 vaccines-12-00583-t002:** Summary of key information from published studies that have identified type I interferon (IFN)-related biomarkers as predictive of vaccine efficacy.

Model	Analytical Method	Vaccine	Main Investigated Issue	Identified Type I IFN-Related Biomarkers	Timing Used for the Expression Analysis & Kinetics	Reference
Whole blood from 40 subjects vaccinated with the YF17D vaccine	Microarray gene expression analysis and profiling using the open source Bioconductor platform (using the R software package) and CRAN package fastICA	Live attenuated yellow fever vaccine (YF17D)	Identification of immune response signatures to yellow fever vaccine	*OAS1*, *OAS2*, *OAS3*, *MX1*, *MX2*, *IFI27*, *IFI30*, *IFI16*, *STAT1* and *IRF7*	Gene expression analysis was performed on the day of vaccination and on days 3, 7, 10, 14, 28 and 60 after vaccination. Overexpression peaked on day 7 after vaccination.	[22]
-PBMCs isolated from the blood of 15 young adult vaccinees (18–45 years old) -Human PBMCs stimulated for 3 h or 12 h with the YF17D vaccine	Microarray gene expression and gene ontology analysis and profiling using hierarchical clustering, the DAVID bioinformatics database and the ingenuity pathway analysis software database	Live attenuated yellow fever vaccine (YF17D)	Identification of early gene signatures that predict adaptive immune responses after vaccination with yellow fever vaccine	*OAS1*, *OAS2*, *OAS3*, *OASL*, *MX1*, *IFIH1*, *IRF7*, *ISG15* and *STAT1*	Gene expression analysis was performed on the day of vaccination and on days 1, 3, 7 and 21 after vaccination. Overexpression peaked on day 7 after vaccination.	[16]
PBMCs isolated from the blood of 19 vaccinated children (6–35 months old)	Microarray gene expression analysis and profiling using the DAVID bioinformatics database and ingenuity pathway analysis	Inactivated enterovirus type 71 (EV71) whole-virus vaccine	Comparison between primary and recall immune responses	*IRF7*, *ISG15*, *MX1*, *IFI27* and *IFIH1*	Gene expression analysis was performed on the day of vaccination and on days 56 and 180 after vaccination.	[23]
Whole blood from healthy adult vaccinees (18 to 64 years old)	Microarray gene expression analysis and profiling using hierarchical clustering and ingenuity pathway analysis	Trivalent inactivated 2009–2010 seasonal influenza vaccine and pneumococcal vaccine (polysaccharide extracts from 23 serotypes)	Comparison between transcriptional signatures of a seasonal influenza vaccine and a 23-valent pneumococcal vaccine in blood	*OAS1*, *OAS2*, *OAS3*, *OASL*, *IF135*, *IRF7*, *IFIH1*, *IFIT1*, *IFIT2*, *IFIT3*, *IFITM1*, *IFITM3*, *MX1*, *IRF9*, *IFI44L*, *ISG15*, *IFI16*, *IFI44l* and *STAT1* (only influenza)	Gene expression analysis was performed on the day of vaccination and on days 1, 3, 7, 10, 14, 21 and 28 after vaccination. Overexpression peaked at 24 h after vaccination for both vaccines and, more consistently, at day 10 after vaccination for the pneumococcal vaccine.	[20]
PBMCs from the blood of 119 healthy male adult vaccinees (18–40 years old)	Microarray gene expression and gene ontology analysis and profiling using hierarchical clustering, the DAVID bioinformatics database and the ingenuity pathway analysis software database	Trivalent inactivated influenza vaccine (TIV)	Analysis of the relationship between gene expression patterns and humoral immune response to vaccination	*STAT1*, *IF135*, *IRF7*, *IFIT1*, *MX1* and *IRF9*	Gene expression analysis was performed on the day of vaccination and on days 1, 3 and 14 after vaccination. Overexpression occurred within 24 h of vaccination.	[19]
-PBMCs isolated from the blood of 56 young adult vaccinees (18–50 years old) -Human PBMCs stimulated for 24 h with the vaccine	Microarray gene expression analysis and profiling using the ingenuity pathway analysis software database	Live attenuated influenza vaccine (LAIV) and trivalent inactivated influenza vaccine (TIV)	Characterization of innate and adaptive responses to different influenza vaccines, with the aim of identifying early molecular signatures that predict the later immune responses	*OAS1*, *OAS2*, *OAS3*, *MX1*, *Mx2*, *IRF7* and *STAT1* (only LAIV)	Gene expression analysis was performed before vaccination and on days 3 and 7 after vaccination. Overexpression peaked on day 3 after vaccination.	[17]
Blood from 90 children and adults (14–24 months old) vaccinated with the influenza vaccine	Microarray gene expression analysis and profiling using hierarchical clustering	Trivalent inactivated influenza vaccine (TIV) with and without the MF59 adjuvant	To show the potential utility of using systems approaches to delineate mechanisms of vaccine immunity and identify correlates of vaccine immunity in children	*OAS1*, *OAS3*, *IFIT1*, *IRF7* and *IFIH1*	Gene expression analysis was performed before vaccination and on days 1, 3, 7 and 28 after vaccination.	[109]
Whole blood from 85 infants (12–35 months old) vaccinated with influenza vaccine by intranasal (LAIV) or intramuscular (TIV) injection	Microarray gene expression analysis, ontology analysis and profiling by using hierarchical clustering (using the R package pvclust) and the ingenuity pathway analysis software database	Live attenuated influenza vaccine (LAIV) and trivalent inactivated influenza vaccine (TIV)	Identification of significantly differentially expressed genes after vaccination with LAIV and TIV influenza vaccines	*IRF7*, *IFIT1*, *IFIT2*, *IFIT3*, *OAS1*, *OAS2*, *OAS3*, *MX1*, *MX2* and *IFI44* (only LAIV)	Gene expression analysis was performed before vaccination and on days 7 to 10 after vaccination. Overexpression peaked on days 7–10 after vaccination.	[110]
Whole blood from 37 vaccinated infants and children (6 months to 14 years old)	Microarray gene expression analysis and profiling using GeneSpring GX 7.3 software and hierarchical clustering	Trivalent inactivated influenza vaccine (TIV) or live attenuated influenza vaccine (LAIV)	Comparing parameters of immune responses elicited by TIV and LAIV vaccines in children to determine whether the early changes in the expression of certain immune related genes correlates with the antibody responses	*OASL*, *OAS3*, *IFIT1*, *IFIT3*, *IFI44L* and *ISG15*	Gene expression analysis was performed before vaccination and on days 1, 7 and 30 after vaccination. Overexpression occurred on day 1 after vaccination.	[31]
Whole blood and serum from 60 healthy adults (18–45 years old) enrolled in a randomized phase I/II clinical trial and vaccinated by transcutaneous, intradermal or intramuscular routes	Microarray gene expression analysis and profiling by hierarchical clustering, ingenuity pathway analysis and logistic regression analysis	Trivalent influenza vaccine (TIV, season 2012–2013, 1:1:1 ratio)	Comparison of the immunogenicity of a seasonal influenza vaccine after administration by different novel immunization routes to discover early biomarkers of the immune response and explore any possible influence of the administration routes	*STAT1*, *IRF9*, *IFI35* and *IRF7*	Gene expression analysis was performed before vaccination and on day 1 after vaccination. Overexpression peaked at 24 h after vaccination.	[18]
Blood from 119 healthy male adult vaccinees (18–40 years old)	Microarray gene expression analysis and integrative genomic analysis and profiling using the DAVID bioinformatics database and ingenuity pathway analysis	Trivalent influenza vaccine (TIV)	Identification of genetic factors that influence responsiveness and immunity to a seasonal influenza vaccine in healthy adults	*OAS1*	Gene expression analysis was performed before vaccination and on days 1, 3 and 14 after vaccination	[111]

**Table 3 vaccines-12-00583-t003:** Summary of key information from studies that have identified type I interferon (IFN)-related biomarkers as predictive of vaccine safety.

Model	Analytical Method	Vaccine	Main Investigated Issue	Conventional Animal Tests	Identified Type I IFN-Related Biomarkers	Reference
Lungs isolated from rats intraperitoneally injected with the pertussis vaccine	DNA microarray gene expression analysis, QuantiGene Plex (QGP) assay and profiling by two-dimensional hierarchical clustering	Pertussis vaccine and toxicity reference pertussis vaccine (whole-cell inactivated pertussis vaccine)	Identification of toxicity-related biomarkers of the pertussis vaccine	ATT	*MX2*, *IRF7* and *IFI27L*	[24]
Lungs isolated from rats intraperitoneally injected with the influenza vaccine	DNA microarray gene expression analysis and profiling by hierarchical cluster analysis	Inactivated monovalent A/H5N1 whole-virion influenza vaccine adjuvanted with aluminum hydroxide (PDv), inactivated whole trivalent influenza vaccine (WPv) and trivalent HA influenza vaccine (HAv)	Comparison between quality of different influenza vaccines (PDv, WPv and HAv)	ATT and LTT	*MX1*, *MX2*, *IRF7*, *IFI47*, *IFRD1* and *FLN29*	[25]
Lungs isolated from rats intraperitoneally injected with the influenza vaccine	QuantiGene Plex (QGP) technology and hybridization with branched DNA (bDNA) amplifier combined with multi-analyte magnetic beads and profiling by hierarchical cluster analysis	Inactivated monovalent A/H5N1 whole-virion influenza vaccine adjuvanted with aluminum hydroxide (PDv), inactivated whole trivalent influenza vaccine (WPv) and trivalent HA influenza vaccine (HAV)	Comparison between the quality of influenza vaccine batches from different manufacturers	ATT	*MX1*, *IRF7*, *IFI47*, *TRAFD1* and *IFRD1*	[26]
Lungs isolated from rats intraperitoneally injected with the influenza vaccine	QuantiGene Plex (QGP) technology and hybridization with branched DNA (bDNA) amplifier combined with multi-analyte magnetic beads and profiling by hierarchical cluster analysis	Inactivated influenza virus vaccine used as a toxicity reference vaccine and concentrated bulk materials of influenza HA vaccine (HA-A to HA-D)	Comparison between biomarker evaluation and animal testing for dose-dependent behavior to show that animal testing does not produce dose-dependent results in serially diluted bulk materials of influenza HA vaccines	ATT and LTT	*MX2*, *IRF7*, *IFI47* and *TRAFD1*	[27]
Lungs isolated from mice injected with the influenza vaccine	QuantiGene Plex (QGP) assay and multiple regression analysis	Toxicity reference influenza vaccine (whole-virion inactivated influenza virus) and influenza hemagglutinin split vaccine (HAv)	Selection of potential biomarkers related to leukocyte reduction from the set of biomarkers	LTT	*MX2*, *TRAFD1*, *IRF7*, *IFI47* and *IFRD1*	[28]
Lungs isolated from mice after intranasal administration of the influenza vaccine	QuantiGene Plex (QGP) assay and multiple regression analysis	Toxicity reference influenza vaccine (whole-virion inactivated influenza virus) and influenza hemagglutinin split vaccine (HAv)	To demonstrate the utility of the biomarker gene set for the safety evaluation of the nasal route for influenza vaccine administration	ATT and LTT	*MX2*, *IRF7*, *IFI47*, *TRAFD1* and *IFRD1*	[29]
Lungs isolated from vaccine-treated mice after intramuscular, intraperitoneal and nasal administration	QuantiGene Plex (QGP) assay, logistic regression analyses and profiling by hierarchical clustering analyses	Subvirion influenza vaccine (HAv), poly I:C adjuvanted hemagglutinin split vaccine HAv, AddaVax™-adjuvanted HAv (squalene-based oil-in-water emulsion adjuvant similar to MF59, which is a safe and potent adjuvant for the use with human vaccines), non-adjuvanted hemagglutinin split vaccine and toxicity reference vaccine (whole-virion inactivated influenza vaccine)	To develop a practical system for the safety assessment of different vaccine and adjuvant inoculation routes	ATT and LTT	*TRAFD1*, *IRF7*, *MX2* and *IFI47*	[30]
Lungs isolated from humanized mice (NOG mice are intravenously engrafted with human PBMCs) injected with the influenza vaccine	QuantiGene Plex (QGP) assay, flow cytometric analysis and multiple regression analysis	Hemagglutinin split vaccine (HAv) and toxicity reference influenza vaccine (whole-virion inactivated influenza vaccine)	To develop a novel humanized mouse model retaining human innate immunity-related cells to assess the safety of influenza vaccines using the (17–20) biomarker gene set	LTT	*MX2* and *TRAFD1* (not significant)	[116]
Donor-derived human PBMCs stimulated for 16 h with different doses of influenza vaccine	QuantiGene Plex (QGP) assay, flow cytometric analysis and multiple regression analysis	Hemagglutinin split vaccine (HAv) and toxicity reference influenza vaccine (whole-virion inactivated influenza vaccine)	To confirm the utility of the (17–20)-biomarker gene set for the safety evaluation of vaccines in humans and to develop a novel in vitro system for the safety evaluation of vaccines using human samples	-	*MX2*, *IRF7*, *TRAFD1* and *TRAFD1*	[117]

## Data Availability

All data related to this study are presented and published here.

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
