# Peer review of "Feasibility of Using a Type I IFN-Based Non-Animal Approach to Predict Vaccine Efficacy and Safety Profiles"

_vaccines, 2024, doi:10.3390/vaccines12060583_

Round 1

Reviewer 1 Report (Previous Reviewer 2)

Comments and Suggestions for Authors

The revised version of Abdel-Haq's review manuscript incorporates newly added evidence and information, however, there remain significant issues to be addressed:

1. The title emphasizes biomarkers based on "non-animal methods" to assess vaccine efficacy and safety, however, critical references cited by the author (e.g., References 29-31) all involve animal models for vaccination and immunity assessment. Consequently, the relevance of the "non-animal approach" appears questionable, decreasing the manuscript's novelty and importance. It is imperative to acknowledge that antigen-induced specific immune responses serve as more standard and essential biomarkers compared to general type I IFN-mediated in vivo activation.

2. Sections 4 and 5 underscore the potential limitations of utilizing type I IFN as a biomarker. Indeed, certain type I IFN-triggered cytokines may contribute to vaccine-induced antibody and cellular responses, but the evaluation of vaccine efficacy and safety relies solely on a few key cytokines is questionable. This limited assessment does not guarantee the vaccine's ability to elicit an antigen-specific immune response. Thus, the relevance of type I IFN-related cytokines (e.g., MX2 and IRF7) appears diminished for vaccinologists and researchers.

3. Despite the addition of new content and information (totaling 38 pages), the current manuscript structure lacks well-organization, characterized by an abundance of small paragraphs. Additionally, the author fails to effectively summarize key references within the main text, and in the Discussion section the author mainly provides current evidence or cited reports, lacking personal discussion and opinions. Consequently, the overall quality of this manuscript has not substantially improved compared to the initial submitted version.

Author Response

The revised version of Abdel-Haq's review manuscript incorporates newly added evidence and information, however, there remain significant issues to be addressed:

1. The title emphasizes biomarkers based on "non-animal methods" to assess vaccine efficacy and safety, however, critical references cited by the author (e.g., References 29-31) all involve animal models for vaccination and immunity assessment. Consequently, the relevance of the "non-animal approach" appears questionable, decreasing the manuscript's novelty and importance. It is imperative to acknowledge that antigen-induced specific immune responses serve as more standard and essential biomarkers compared to general type I IFN-mediated in vivo activation.

Response Thank you for this observation. However, to set up a new biomarkers-based approach for vaccine assessment in non-animal methods, the biomarkers that are potentially able to predict the immunogenicity (in terms of antibody titer, or B and T cells responses) or reactogenicity (adverse effects and/or toxicity) of the tested vaccines must be first identified and selected in studies performed in vivo. Please see lines 858-864 of the manuscript (The transcriptomic approach, using in vitro and ex vivo models or blood samples from vaccinated human subjects, allows researchers to predict vaccines efficacy and safety. This was possible through the identification of vaccine-associated immune signatures whose expression level was significantly correlated with the magnitude of vaccine immunogenicity (vaccine-related neutralizing antibody and antigen-specific CD8+ T cells responses) and/or reactogenicity (vaccine-related body weight loss and reduction of blood leukocytes) determined in in vivo measurements). Thus, the biomarkers whose expression level was significantly correlated with the vaccine immune response or toxicity determined in the in vivo bioassay or in human subjects were selected. Only after demonstrating the reliability and robustness of the selected biomarkers and their equivalent performance to animal testing, they can be considered qualified for future use in animal-free methods to assess vaccines. It is worth noting that only vaccines with well-characterized efficacy or toxicity profiles were used in the studies analysed in this article, and therefore, the expression levels of the identified biomarkers can be considered correlated with the efficacy or toxicity of the tested vaccines. For more information about potential biomarkers other than Type I IFNs, please also see perspective section, which describes and suggests other potential biomarkers for vaccine evaluation in animal-free models, possibly in the future.

2. Sections 4 and 5 underscore the potential limitations of utilizing type I IFN as a biomarker. Indeed, certain type I IFN-triggered cytokines may contribute to vaccine-induced antibody and cellular responses, but the evaluation of vaccine efficacy and safety relies solely on a few key cytokines is questionable. This limited assessment does not guarantee the vaccine's ability to elicit an antigen-specific immune response. Thus, the relevance of type I IFN-related cytokines (e.g., MX2 and IRF7) appears diminished for vaccinologists and researchers.

Response Sections 4 is referred to limitations concerning the use of non-animal methods “Potential limitations that may hamper vaccine characterization using alternative non-animal models”, while section 5 underscores some potential critical aspects that may influence the quantity and quality of biomarker. All new tests, assays and approaches present potential limitations and challenges, which should never discourage from improving and advancing the field of biomarker research. For some potential limitations and critical issues highlighted, solutions have been suggested in this manuscript.

3. Despite the addition of new content and information (totaling 38 pages), the current manuscript structure lacks well-organization, characterized by an abundance of small paragraphs. Additionally, the author fails to effectively summarize key references within the main text, and in the Discussion section the author mainly provides current evidence or cited reports, lacking personal discussion and opinions. Consequently, the overall quality of this manuscript has not substantially improved compared to the initial submitted version.

Response I apologize to the reviewer, I really do not understand the significance of this comment “the current manuscript structure lacks well-organization, characterized by an abundance of small paragraphs”? I really apologize because the structure has already been revised based on the comments of the other reviewers and has also been judged adequate. Regarding the following comment, “Additionally, the author fails to effectively summarize key references within the main text, and in the Discussion section the author mainly provides current evidence or cited reports, lacking personal discussion and opinions”, the reviewer had the opportunity to make this and other observations in the first round of review but made no such comments, although the main text has not changed with respect to before. I therefore sincerely apologize but I strongly disagree with the reviewer’s affirmation because the main text has been carefully prepared, and the references have been accurately selected. Also, this section has been carefully analysed and discussed within the same sections and in the discussion. However, any suggestion on key references is gladly acceptable. In the discussion section, the author expressed several personal scientific opinions also through the literature data. This is the authors’ own style, which is obviously different from that of the reviewer. If the author does not say “in my opinion”, does that mean the author expressed no opinion? Thank you.

Reviewer 2 Report (Previous Reviewer 3)

Comments and Suggestions for Authors

The author has modifed the manuscript significanlty, and the author had replied effieicently to my comments which raised in the previous submission. 

Comments on the Quality of English Language

langugae is ok

Author Response

The author has modifed the manuscript significanlty, and the author had replied effieicently to my comments which raised in the previous submission. Response Thank you very much.

Reviewer 3 Report (Previous Reviewer 4)

Comments and Suggestions for Authors

The manuscript is now more balanced after revision. While there are still doubts whether interferon modulation can help vaccine development, this review does provide some insights into how interferon can be considered based on specific situations. However, Table 1 is currently over-simplistic and should be removed. For instance, the claims that in vitro is more representative than animal models is incorrect as this is not always the case.

Author Response

The manuscript is now more balanced after revision. While there are still doubts whether interferon modulation can help vaccine development, this review does provide some insights into how interferon can be considered based on specific situations. However, Table 1 is currently over-simplistic and should be removed. For instance, the claims that in vitro is more representative than animal models is incorrect as this is not always the case. Response - Thank you for your comment. Table 1 was added after a comment and an explicit request from one of the reviewers. Therefore, removing or keeping this table is a decision that I would like to entrust to the Editor. In my opinion, table 1 only displays a summary of what the literature reports, Therefore, this table does not add information but also does not affect the quality of the manuscript. It may instead aid the reader focus on the advantage of moving towards non-animal models and/or improving the performance of animal models. - I agree with the reviewer that the in vitro experimentations are not representative than the animal model in all cases. This is not confirmed in the current manuscript, which instead encourages the reduction of animal use, which is possible. The gradual replacement of the animal use with alternative methods is also encouraged when this is possible, but not in all cases. In support of this, section 4 (which discusses potential limitation deriving from in vitro methods) highlights the need to find alternative methods more relevant than the methods used in in vitro models properly because they are not always able to reflect the in vivo situation. Thank you

This manuscript is a resubmission of an earlier submission. The following is a list of the peer review reports and author responses from that submission.

Round 1

Reviewer 1 Report

Comments and Suggestions for Authors

There are many kinds of vaccines, including live attenuated vaccines, subunit vaccines, DNA vaccines, mRNA vaccines, etc. One or more adjuvants may be used for each vaccine. Both vaccines and adjuvants can execute different effects on IFN production and IFN signaling for producing ISGs. Although the author offers a "bright" path, it has been very difficult to establish this system until now.

For example, live attenuated vaccines must protect the immmnuized ainmals  and clear the live virus or maintain a minimum viral load. 

DNA or mRNA vaccines can produceIFNs and ISGs, but the primary goal of immunity is to produce protective antibodies. However,there was barely correlation between IFNs and ISGs induced by DNA or mRNA vaccines and the protective effect.

Most subunit vaccines, which only include proteins (antigens) with different adjuvants, can produce neutralizing antibodies without IFNs and ISGs (if have , may be consided as bad responses), how to evaluate them?

Comments on the Quality of English Language

The English language is fine. 

Reviewer 2 Report

Comments and Suggestions for Authors

The review manuscript by Abdel-Haq highlights a non-animal approach based on the type I IFN signal pathway that might serve as an alternative for predicting vaccine safety and efficacy. Although this review carefully summarizes significant relevant evidence and information, there are some important concerns to be addressed.

1. Type I IFNs are known to play a role in both innate and adaptive immune cells upon pathogenic infection and vaccination. However, the production and levels of the type I IFN response can be influenced by various factors, including the type of vaccine (vaccine platform), adjuvant usage and dosage, and administration routes, in addition to the vaccine itself. Particularly, the current mRNA vaccine platform is mainly developed using modified nucleotides to reduce the type I IFN response, which has been reported to negatively affect vaccine efficacy. Moreover, the timing of the type I IFN response in vivo is significant in terms of vaccine efficacy. Thus, as the author mentioned the potential limitations, it would be quite challenging to conclude that several identified type I IFN-mediated cytokines, using in vitro assays, can serve as biomarkers for assessing vaccine safety and efficacy. 

2. In the second section on "interferon (IFN)," most of the content is not novel, as numerous literatures and reviews have covered this topic, potentially reducing the readership of this review.

3. In the third section, as a crucial part of the manuscript, the author did not organize the structure well. Key information, discussion generation, and the summary are not well-presented. Additionally, the author did not clearly and comprehensively mention the in vitro cell lines, assays, and settings that can be used to assess vaccine efficacy (besides innate immune response, e.g., antibody response or cellular response) and safety in detail.

4. The current tables mainly focus on live attenuated pathogenic vaccines or inactivated vaccines. The author should put more effort into collecting information on whether IFN-mediated biomarkers are also suitable for other vaccine platforms.

5. Importantly, are there known cytokines that might be detrimental to reducing vaccine efficacy and safety? These mediators should be excluded as biomarkers.

6. Small remarks include typos in the manuscript (e.g., Line 858).

Reviewer 3 Report

Comments and Suggestions for Authors

It is an interesting review and comperhensive one that describes the in vitro models used for evaluation of vaccine efficacy and safety as an alternative model to the animal model.

The review includes the following points: a) Interfron types and siganling pathways.  b) Interferon stimulating genes.  c)  Type I interferon (IFN)-related biomarkers as predictive of  the vaccine efficacy.   d)  limitations of the in vitro model for vaccine evaulation.

Some points need to be added to improve the quality of manuscript:

a) since vaccine evaluation is based mainly on the adaptive immune responses (Ab and T cells), therefore, the author should provide rationale of the review  focuing on interferon

b) Please mention that limitations of use animals as a model for vaccine evaluation. And why the researchers search for in vitro model as alternative to animal model.

c) The development of 3D organoid model as an acceptable model for drug and vaccine evaulation 

d) Please add table about the compersion between animal and non-animal model regarding their efficeincies  in vaccine evaluation.

e) The author should provide the signaling link between IFN and Ab production and/or T cells 

Comments on the Quality of English Language

moderate language editing

Reviewer 4 Report

Comments and Suggestions for Authors

In this review, Hanin Abdel-Haq explore the possibility of using interferon-related genes as indicators of vaccine safety and efficacy. While it is tempting to replace in vivo studies with in vitro studies, this article is over-simplistic, which may misguide readers to think that measuring interferon-related genes would be sufficient to guide vaccine safety and efficacy. My comments are as follows:

1. While it is logical to think that interferon responses are associated with innate immunity that is important for adaptive immune responses, it is also important to note that interferon responses are also associated with more side effects from vaccines. Studies looking into severe adverse events have documented increased levels of interferon, which suggest that this cannot be a solution to solving the question for safety and efficacy. In addition, in the study by Chan et al., JCI Insight, 2017, the interferon genes that are correlated with immunogenicity are also correlated with adverse event outcome, so evaluation by interferon is going to be a tricky.

2. What levels of interferon induction would one consider as a good vaccine? And what levels are informative of serious adverse events? We have no answers to any of these questions at present. The author did not provide a good perspective on this critical question as well.

3. In vitro studies cannot inform on the physiological effects caused by vaccines, so the safety assessment for in vitro studies will likely be limited. The author also did not mention how the interferon studies can be supplemented to make them more interpretable

4. The author argue that interferon responses are associated with better vaccine immunogenicity. While it is true that the interferon responses are critical to induce innate immune responses, as well as to suppress viral replication to make them attenuated, the recent study by Thomas Hagan et al., Nature Immunology, 2022, analysing 13 different vaccine responses suggest that the converging signature that are associated with antibody responses are not interferon. Instead, the converging signature was plasma cell and immunoglobulin. Thus, interferon measurements cannot be the only read-out to assess vaccine immunogenicity. The underlying mechanisms are likely more complex.

5. The other issue of using interferon as read-out is related to the cell types used for measuring interferon responses. Different cell types have different propensity to induce interferon, so this need to be explained and extended in great detail.